# ZNF516 suppresses *EGFR* by targeting the CtBP/LSD1/CoREST complex to chromatin

Lifang Li[1], Xinhua Liu[2], Lin He[1], Jianguo Yang[1], Fei Pei[3], Wanjin Li[1], Shumeng Liu[1], Zhe Chen[1], Guojia Xie[1], Bosen Xu[1], Xia Ting[1], Zihan Zhang[1], Tong Jin[1], Xujun Liu[1], Wenting Zhang[1], Shuai Yuan[1], Ziran Yang[1], Chongyang Wu[1], Yu Zhang[1], Xiaohan Yang[1], Xia Yi[1], Jing Liang[1], Yongfeng Shang[1,2] & Luyang Sun [1]

EGFR is required for animal development, and dysregulation of EGFR is critically implicated in malignant transformation. However, the molecular mechanism underlying the regulation of EGFR expression remains poorly explored. Here we report that the zinc-finger protein ZNF516 is a transcription repressor. ZNF516 is physically associated with the CtBP/LSD1/CoREST complex and transcriptionally represses a cohort of genes including EGFR that are critically involved in cell proliferation and motility. We demonstrate that the ZNF516–CtBP/LSD1/CoREST complex inhibits the proliferation and invasion of breast cancer cells in vitro and suppresses breast cancer growth and metastasis in vivo. Significantly, low expression of ZNF516 is positively associated with advanced pathological staging and poor survival of breast carcinomas. Our data indicate that ZNF516 is a transcription repressor and a potential suppressor of EGFR, adding to the understanding of EGFR-related breast carcinogenesis and supporting the pursuit of ZNF516 as a potential therapeutic target for breast cancer.

[1] Key Laboratory of Carcinogenesis and Translational Research (Ministry of Education), Beijing Key Laboratory of Protein Posttranslational Modifications and Cell Function, Department of Biochemistry and Molecular Biology, School of Basic Medical Sciences, Peking University Health Science Center, Beijing 100191, China. [2] Department of Biochemistry and Molecular Biology, School of Basic Medical Sciences, Tianjin Medical University, Tianjin 300070, China. [3] Department of Pathology, School of Basic Medical Sciences, Peking University Health Science Center, Beijing 100191, China. Correspondence and requests for materials should be addressed to L.S. (email: luyang_sun@hsc.pku.edu.cn)

Epidermal growth factor receptor (EGFR) is a transmembrane glycoprotein composed of an extracellular ligand-binding domain, a single membrane-spanning region, a juxta membrane nuclear localization signal (NLS), a tyrosine kinase domain, and a tyrosine-rich C-terminal tail[1]. As the identification of a link between *EGFR* and the transforming viral oncogene *v-erb-B*[2], it has been well-established that EGFR is involved in malignant transformation and progression of a broad variety of cancers[3–5]. Indeed, EGFR overexpression has been reported in cancers originating from bladder, brain, breast, cervical, uterine, colon, esophageal, glioma, lung, ovarian, pancreatic, and renal cell[3–7], and this dysregulation is often associated with a more aggressive phenotype and accordingly worse survival of the cancer patients[8]. This scenario makes the EGFR family an ideal target to be exploited for cancer therapeutics[9–11]. Nevertheless, therapeutic agents are ultimately limited by the emergence of secondary EGFR mutations and other molecular mechanisms that confer drug resistance[12, 13].

Despite the extensive molecular and functional characterization of EGFR and a continuing effort in pursuing anti-EGFR cancer therapies, the molecular mechanism underlying the regulation/dysregulation of EGFR expression remains poorly explored. This issue is of particular importance as it was noted that amplifications in the *EGFR* gene are restricted to regions of the regulatory sequence in the 5′-end of intron 1 and associated with EGFR expression in epithelial breast tumors[14], implying the importance of transcriptional regulation of EGFR in breast carcinogenesis.

Zinc-finger protein 516 (ZNF516) (KIAA0222) is a member of the Krüppel C2H2-type zinc-finger protein family[15]. It has been reported that ZNF516 has an important role in Dupuytren's contracture (DC) development, thus is considered as a candidate of molecular targets for treating DC[16]. ZNF516 have been implicated in congenital vertical talus[17] and reported to influence bone mineral density[18]. Znf516 null mice die immediately after birth due to a yet-to-be-defined role during development[19]. At the molecular level, it is shown that Znf516 is a cold-inducible factor capable of activating UCP1 or PGC1α transcription, thereby promoting browning of white fat and development of brown fat in mice[19, 20]. However, several studies suggest that ZNF516 is implicated in transcription repression[21–24]. Dysfunction of ZNF516 has been implicated in various pathological states including malignancies. It is reported that *ZNF516* is subject to frequent copy number loss that is associated with chromosomal instability and aneuploidy onset at adenoma–carcinoma transition in colorectal cancer[25], and hypermethylation on *ZNF516* promoter is considered as a better biomarker for cervical neoplasia[26]. However, the molecular mechanism underlying the role of ZNF516 in tumorigenesis is still poorly understood.

C-terminal binding protein (CtBP) was originally identified by its interaction with the C terminus of adenovirus E1a protein and its ability to negatively regulate oncogenic transformation[27, 28]. In effect, CtBP forms heterodimer/homodimer in the presence of nicotinamide adenine dinucleotide[29], thereby repressing gene transcription through recruitment of epigenetic modifiers including histone deacetylases (HDAC1 and HDAC2), histone methyltransferases (G9a and GLP), and histone demethylase (LSD1)[15, 23, 30, 31]. In addition, corepressor of RE1 silencing transcription factor (CoREST) is frequently found in this complex[32, 33]. It is believed that CtBP itself is not capable of binding DNA; it needs to be recruited to promoter elements of specific genes by interacting with chromatin targeting/DNA-binding transcription factors possessing a classical Pro-X-Asp-Leu-Ser (PXDLS) and/or Arg-Arg-Thr (RRT) motif[15, 34, 35]. Consequently, it is proposed that CtBP acts to bridge a particular transcription factor, such as ZEB1/2 and ZNF217, and its recruited

corepressor complex[36, 37]. Biologically, it has been reported that CtBP functions as either tumor suppressor or promoter, depending on the context of its associated partners[38–41].

In this study, we report that ZNF516 functions as a transcription repressor. ZNF516 is physically associated with the CtBP/LSD1/CoREST corepressor complex and transcriptionally represses EGFR expression. We demonstrate that the ZNF516 inhibits the proliferation and invasive potential of breast cancer cells in vitro and suppresses breast cancer growth and metastasis in vivo. We explore the clinical significance of the ZNF516–CtBP/LSD1/CoREST–EGFR axis in breast carcinomas.

## Results

**ZNF516 is a transcription repressor.** In an effort to explore the mechanistic role of ZNF516 in breast cancer carcinogenesis, we cloned the gene encoding for ZNF516 from a human mammary cDNA library. *ZNF516* is mapped to chromosome 18q23 and consists of eight exons and seven introns. The predicted molecular weight of ZNF516 is 124.3 kDa. Bioinformatics analysis indicates that ZNF516 harbors 10 C2H2-type zinc fingers (Supplementary Fig. 1a). Amino-acid sequence alignment reveals that the similarity of human ZNF516 with homologs in other organisms is 98.3% in *Pan troglodytes*, 80.3% in *Mus musculus*, 55.6% in *Gallus gallus*, 47.2% in *Xenopus tropicalis* and 26.4% in *Danio rerio* (Supplementary Fig. 1b). Phylogenetic analysis also indicates that ZNF516 is an evolutionarily well-conserved gene (Supplementary Fig. 1c).

To confirm the expression of ZNF516 protein, FLAG-tagged ZNF516 (FLAG-ZNF516) expression plasmids were transfected into HEK293T and MCF-7 cells. Cellular proteins were extracted from these cells as well as from several other cell lines and analyzed by western blotting with a monoclonal antibody against FLAG or polyclonal antibodies against ZNF516. The results showed that endogenous ZNF516 is a protein with a molecular weight of ~140 kDa, and that ZNF516 is expressed at variable levels in different cell lines (Fig. 1a). Immunofluorescent imaging of ZNF516 in MCF-7 and MDA-MB-231 cells indicate that ZNF516 is primarily localized in the nucleus (Fig. 1b).

The Krüppel C2H2-type zinc-finger family of proteins has been implicated in transcriptional regulation[15]. However, as stated above, both activation[19, 20] and repression activity[21–24] have been reported for ZNF516. To determine the transcriptional activity of ZNF516 in our system, we fused the full length of ZNF516 to the C terminus of Gal4 DNA-binding domain and tested the transcriptional activity of the fused construct in HEK293T and MCF-7 cells. We utilized three different Gal4-driven luciferase reporter systems, which differ in basal promoter elements (Fig. 1c). The results showed that ZNF516 dramatically inhibits the reporter activity in a dose-dependent manner in all of the three reporter systems in both HEK293T and MCF-7 cells (Fig. 1d), suggesting that ZNF516 is involved in transcription repression. Meanwhile, overexpression of FLAG-ZNF516 did not affect the activity of Gal4-driven reporter in MCF-7 cells (Fig. 1e), suggesting that ZNF516 must physically bind to DNA to exert its transcriptional activity.

**ZNF516 is physically associated with the CtBP/LSD1/CoREST complex.** To gain a mechanistic insight into the ZNF516-mediated transcription repression, we then employed affinity purification and mass spectrometry to interrogate the interactome of ZNF516 in vivo. HEK293T cells embody some principal attributes for the mass spectrometric analysis of protein interactome, in which the large amounts of protein were required: quick and easy reproduction and maintenance, high efficiency of transfection and protein production; and faithful translation and

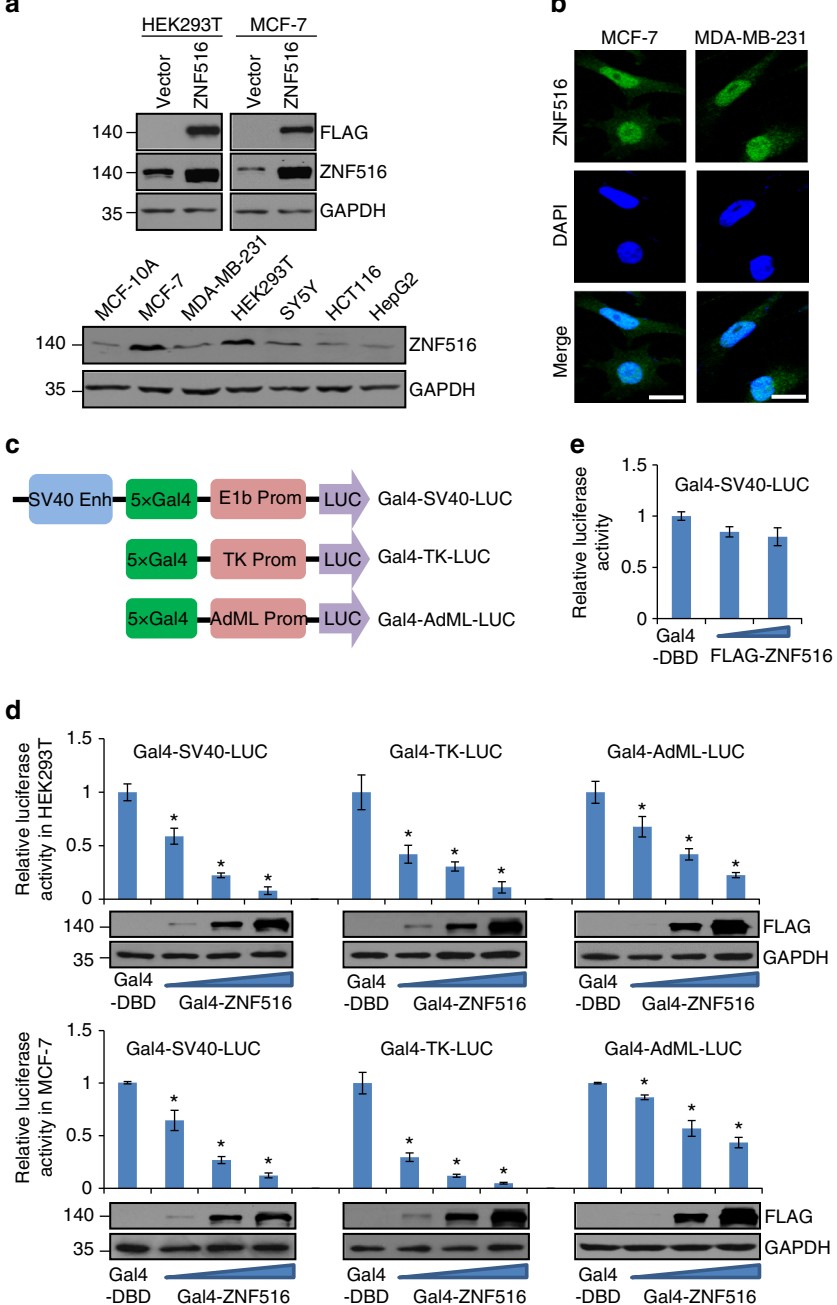

**Fig. 1** ZNF516 is a transcription repressor. **a** Western blotting analysis of ZNF516 protein expression. HEK293T or MCF-7 cells were transfected with empty vector or FLAG-ZNF516. Cellular proteins were extracted from indicated cell lines and western blotting was performed with a monoclonal antibody against FLAG, GAPDH, or polyclonal antibodies against ZNF516. **b** Subcellular localization of ZNF516 protein. The distribution of endogenous ZNF516 was detected by immunofluorescent microscopy with polyclonal antibodies against ZNF516. DAPI staining was included to visualize the cell nucleus. *Scale bar* 20 μm. **c** The schematic diagram showing the Gal4-luciferase reporters. **d** Transcription repression by ZNF516. HEK293T or MCF-7 cells were transfected with different amounts of Gal4-ZNF516 expression plasmids, together with the indicated Gal4-luciferase reporter. Forty-eight hours later, luciferase activity was measured. Relative luciferase activity was calculated as firefly luciferase activity divided by renilla luciferase activity and shown relative to the control (transfected with Gal4-DBD vector). The expression of Gal4-ZNF516 was shown by western blotting. **e** MCF-7 cells were transfected with different amounts of FLAG-ZNF516 plasmids, together with the Gal4-SV40-LUC luciferase reporter. Forty-eight hours later, luciferase activity was measured. Relative luciferase activity was calculated as firefly luciferase activity divided by renilla luciferase activity and shown relative to the control. Each *bar* represents the mean ± S.D. for triplicate experiments. *P*-values were determined by Student's *t*-test. *$P < 0.05$

processing of proteins[42]. Thus, in these experiments, FLAG-ZNF516 was stably expressed in HEK293T cells. Whole-cell extracts were prepared and subjected to affinity purification using an anti-FLAG affinity gel. Mass spectrometric analysis indicates that ZNF516 was co-purified with CtBP1/2, CoREST, LSD1, HDAC1/2, and RBBP4/7, all components of the CtBP/

LSD1/CoREST complex (Fig. 2a and Supplementary Data 1). In addition, USP9X, IRS4, ZEB2, and SIRT1 were also identified in the ZNF516-contaning protein complex(es) (Fig. 2a).

To substantiate the above observations, protein extracts from HEK293T cells overexpressing FLAG-ZNF516 were immunoprecipitated with the anti-FLAG followed by immunoblotting with

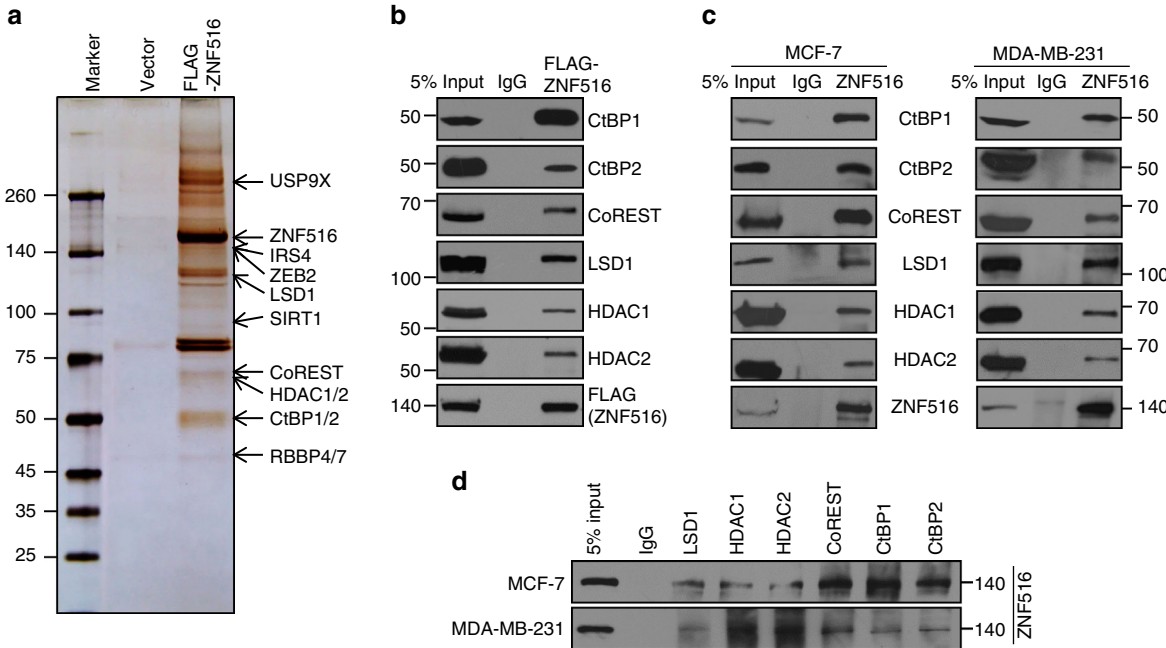

**Fig. 2** Identification of ZNF516-associated proteins. **a** Mass spectrometry analysis of ZNF516-associated proteins. Cellular extracts from HEK293T cells stably expressing FLAG-ZNF516 were immunopurified with anti-FLAG affinity column and eluted with FLAG peptides. The eluates were resolved by SDS-PAGE and silver-stained. The protein bands were retrieved and analyzed by mass spectrometry. **b** Interaction of ZNF516 with the components of the CtBP/LSD1/CoREST complex. Whole-cell lysates from HEK293T cells transfected with FLAG-ZNF516 were prepared and immunoprecipitation was performed with anti-FLAG followed by immunoblotting with antibodies against indicated proteins. **c** Whole-cell lysates from MCF-7 or MDA-MB-231 cells were immunoprecipitated with antibodies against ZNF516 or IgG followed by immunoblotting with the antibodies against the indicated proteins. **d** Whole-cell lysates from MCF-7 or MDA-MB-231 cells were immunoprecipitated with antibodies against indicated proteins or IgG followed by immunoblotting with the antibodies against ZNF516

antibodies against CtBP1/2, CoREST, LSD1, or HDAC1/2. The results confirmed the interaction of ZNF516 with the components of the CtBP/LSD1/CoREST complex (Fig. 2b). In addition, total proteins from MCF-7 and MDA-MB-231 cells were extracted and co-immunoprecipitation experiments were performed with antibodies detecting the endogenous proteins. Immunoprecipitation (IP) with antibodies against ZNF516 followed by immunoblotting (IB) with antibodies against CtBP1/2, CoREST, LSD1, or HDAC1/2 demonstrated that all the tested proteins were efficiently co-immunoprecipitated with ZNF516 (Fig. 2c). Reciprocally, IP with antibodies against components of the CtBP/LSD1/CoREST complex followed by IB with antibodies against ZNF516 also showed that ZNF516 was efficiently co-immunoprecipitated by all the components of the CtBP/LSD1/CoREST complex (Fig. 2d). These results support a notion that ZNF516 is physically associated with the CtBP/LSD1/CoREST complex in vivo.

To further support this deduction, nuclear proteins from MCF-7 cells were fractionated by fast protein liquid chromatography (FPLC) with Superose 6 columns and a high salt extraction and size exclusion approach. We found that the native ZNF516 from MCF-7 cell nuclear extracts was eluted with an apparent molecular mass much greater than that of the monomeric protein; ZNF516 immunoreactivity was detected in chromatographic fractions from the Superose 6 column with a relative symmetric peak centered between ~669 and ~2,000 kDa (Fig. 3a). Importantly, the elution pattern of ZNF516 largely overlapped with that of the CtBP/LSD1/CoREST complex proteins including CtBP1/2, CoREST, LSD1, and HDAC1/2. Significantly, analysis of FLAG-ZNF516 affinity eluate by FPLC after Superose 6 gel filtration in HEK293T cells stably expressing FLAG-ZNF516 detected a multiprotein complex containing

ZNF516, CtBP1/2, CoREST, LSD1, and HDAC1/2 (Fig. 3b). Collectively, these results support the existence of the ZNF516–CtBP/LSD1/CoREST complex in vivo.

To further consolidate the interaction between ZNF516 and the CtBP/LSD1/CoREST complex and to investigate the molecular details involved in this interaction, glutathione S-transferase (GST) pull-down experiments were performed with bacterially expressed GST-fused deletion mutants of ZNF516 (Fig. 3c) and in vitro transcribed/translated individual components of the CtBP/LSD1/CoREST complex. The results revealed that the N-terminal region (1–9 Zinc fingers) of ZNF516 is responsible for its interaction with ELM2-SANT II domain of CoREST, and with the PLDLS and RRT-binding domains of CtBP1/2 (Fig. 3d). No direct interaction of ZNF516 was detected with the other components of the CtBP/LSD1/CoREST complex that we tested (Fig. 3d). Reciprocal GST pull-down experiments with GST-fused CtBP1, CtBP2, CoREST, LSD1, HDAC1, or HDAC2 and in vitro transcribed/translated ZNF516 yielded similar results (Fig. 3e). Altogether, these results further support the specific interaction between ZNF516 and the CtBP/LSD1/CoREST complex.

**Identification of transcriptional targets for ZNF516**. To investigate the functional significance of the physical association between ZNF516 and the CtBP/LSD1/CoREST complex, we first tested whether histone deacetylase or histone lysine demethylase activity is required for ZNF516-mediated gene repression. To this end, the reporter assays were repeated in MCF-7 cells under the presence or absence of trichostatin A (TSA), a specific HDAC inhibitor, or tranylcypromine, a LSD1 inhibitor. The results showed that treatment with either TSA or tranylcypromine could alleviate transcription repression of the reporter activity by ZNF516 (Fig. 4a), supporting the physical association of ZNF516

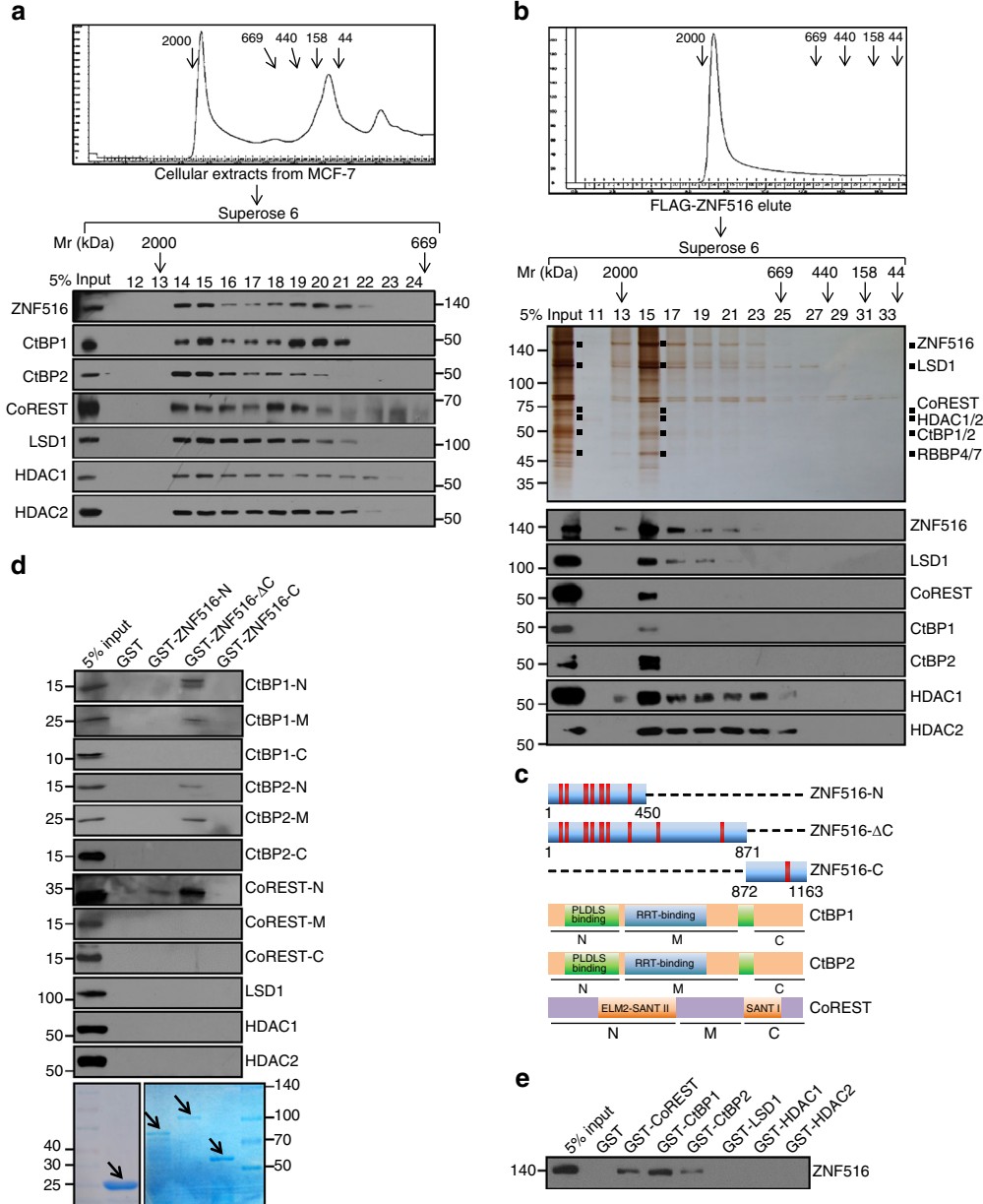

**Fig. 3** ZNF516 interacts with CtBP/LSD1/CoREST complex in vivo and in vitro. **a** Co-fractionation of ZNF516 and CtBP/LSD1/CoREST complex by FPLC. MCF-7 cell proteins were extracted, concentrated, and then fractionated on Superose 6 size exclusion columns. Chromatographic elution profiles and immunoblotting analysis of the chromatographic fractions are shown. The elution positions of calibration proteins with known molecular masses are indicated, and an equal volume from each fraction was analyzed. **b** Silver staining and western blot analysis of the ZNF516-containing complex fractionated by Superose 6 gel filtration. Cellular extracts from HEK293T cells transfected with FLAG-ZNF516 were fractionated on Superose 6 size exclusion columns. Chromatographic elution profiles, silver staining, and western blot analysis of the chromatographic fractions are shown. The elution positions of calibration proteins with known molecular masses are indicated, and an equal volume from each fraction was analyzed. **c** Schematic diagrams of ZNF516, CtBP1/2, and CoREST deletion mutants. **d** GST pull-down experiments were performed with bacterially expressed GST or GST-fused ZNF516 deletion mutants and in vitro transcribed/translated truncation mutants of CtBP1, CtBP2, or CoREST, or full length of LSD1, HDAC1 or HDAC2. Coomassie brilliant blue staining of the GST-fused proteins was shown with *arrows*. **e** Bacterially expressed GST or GST-fused indicated proteins were performed GST pull-down experiments with in vitro transcribed/translated ZNF516

with HDAC1/2 and LSD1, components of the CtBP/LSD1/CoREST complex.

Next, we performed chromatin immunoprecipitation-based deep sequencing (ChIP-seq) to analyze the genome-wide transcriptional targets of ZNF516. In these experiments, ChIP experiments were performed first in MCF-7 cells with antibodies against ZNF516. Following ChIP, ZNF516-associated DNAs were amplified using non-biased conditions, labeled, and then sequenced via HiSeq2500. On the basis of Bowtie Version 2,

33109752 out of 37445556 and 30427894 out of 35785902 reads, were mapped to the human reference genome (GRGH37, hg19) for ZNF516 ChIP and input samples respectively. Using MACS Version2[43] and with a FDR cutoff of 0.05, we identified 14938 ZNF516-specific binding peaks (Fig. 4b). Although ZNF516's association is detected in intronic or intergenic sequences, these sequences are most likely represent enhancer elements, which loop back with promoter to enhance transcription. As transcription factors typically recognize and bind to specific DNA

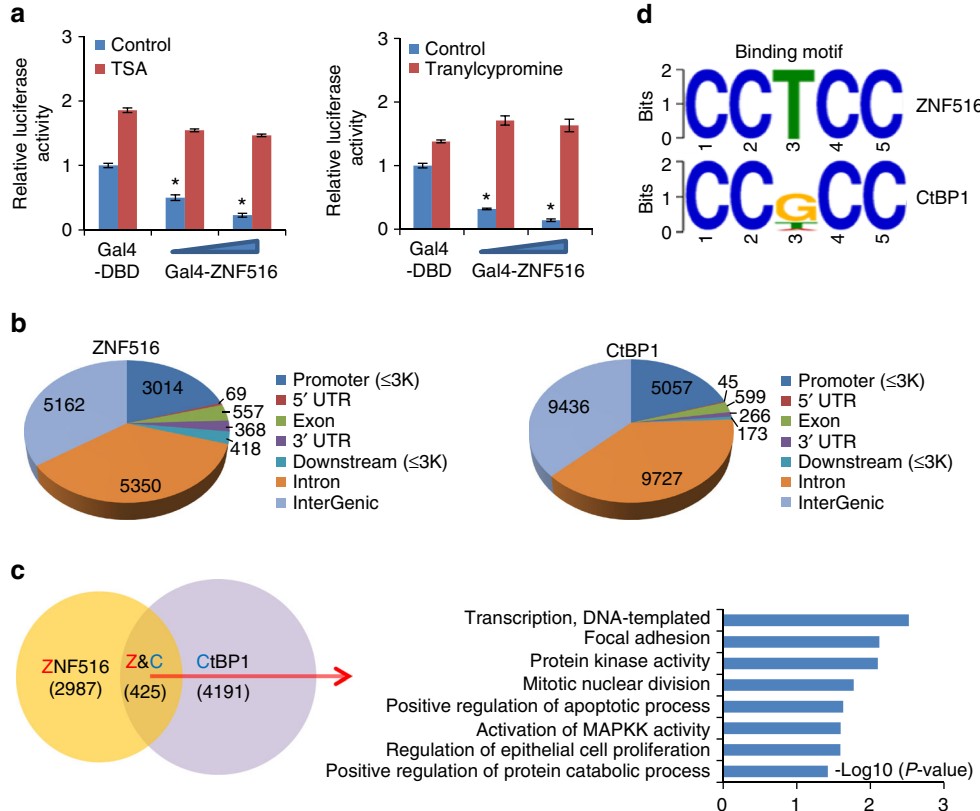

**Fig. 4** Identification of transcriptional targets for ZNF516. **a** MCF-7 cells were transfected with Gal4-DBD or Gal4-ZNF516 plasmids and were treated with trichostatin A (*TSA*) or tranylcypromine. Forty-eight hours later, luciferase activity of Gal4-SV40-LUC was measured. Relative luciferase activity was calculated as firefly luciferase activity divided by renilla luciferase activity and shown relative to the control. Each *bar* represents the mean ± S.D. for triplicate experiments. *P*-values were determined by Student's *t*-test. *P < 0.05. **b** Genomic distribution of ZNF516 and CtBP1 (published in ref. [46]) determined by ChIP-seq analysis in MCF-7 cells. Number of *peaks* in each cluster was indicated. **c** The Venn diagram of overlapping genes targeted by ZNF516 and CtBP1 in MCF-7 cells (*left*). The clustering of the 425 overlapping target genes of ZNF516/CtBP1 into biological process ontologies is shown (*right*). The details of the ChIP-seq experiments were provided in the Materials and Methods and results on the ontologies were provided in Supplementary Data 2. **d** The analysis of ZNF516-bound motifs and CtBP1-bound motifs using MEME suite. Top 10 motifs enriched at ZNF516 peaks were listed in Supplementary Fig. 2

sequences in the promoter of target genes via characteristic DNA-binding domains[44, 45], where transcription initiates, genes in the promoter cluster were then cross-analyzed with published ChIP-seq data for CtBP1 in MCF-7 cells (GEO accession number: GSE36546)[46] for overlapping. A total of 425 genes targeted by ZNF516 and CtBP1 were identified, which were considered to be the targets of the ZNF516–CtBP/LSD1/CoREST complex, and then classified into various cellular biological processes using the Database for Annotation, Visualization and Integrated Discovery (DAVID, https://david.ncifcrf.gov/). These biological processes include focal adhesion, mitotic nuclear division, regulation of epithelial cell proliferation, and apoptosis that are critically involved in cell proliferation and motility (Fig. 4c and Supplementary Data 2). Significantly, analysis of the genomic binding signatures of ZNF516 and CtBP1 revealed indeed a similar binding motif CCTCC, and Find Motif Occurrence (FIMO) scanner[47] identified the 425 genes and 69% of the peaks with this motif (Fig. 4d and Supplementary Fig. 2), supporting the notion that ZNF516 and CtBP1 physically interact and are functionally linked. Quantitative ChIP (qChIP) analysis in MCF-7 cells using specific antibodies against ZNF516, CtBP1, CoREST, or LSD1 on selected genes including *EGFR*, *TGFB3*, *SMAD3*, *BCL3*, *STAT2*, *ERBB3*, *MAP3K13*, *CDKN1A*, *KDM3A*, *TUBB3* that represent each of the classified pathways showed a strong enrichment of ZNF516, CtBP1, CoREST, and LSD1 on the promoters of these genes (Fig. 5a). In addition, measurement of

the mRNA expression of the selected genes by real-time RT-PCR in MCF-7 cells indicates that the expression of these genes decreased when ZNF516 was overexpressed and increased upon ZNF516 knockdown (Fig. 5b). Similar results were obtained in MDA-MB-231 cells (Fig. 5c).

To further support the notion that ZNF516 interacts and recruits the CtBP/LSD1/CoREST complex in transcription repression of target genes, the expression of ZNF516, CtBP1, or CoREST was individually knocked down by their corresponding siRNA in MCF-7 cells. qChIP experiments indicate that depletion of ZNF516, CtBP1, or CoREST resulted in a marked reduction of the recruitment of the corresponding protein at the promoters of the target genes (Fig. 5d). However, while depletion of ZNF516 was associated with a reduced recruitment of CtBP1 and CoREST on the target promoters, depletion of either CtBP1 or CoREST resulted in only marginal or no effect on the recruitment of ZNF516 (Fig. 5d). Consistently, the levels of pan-H3 acetylation (H3Ac) and H3 lysine 4 trimethylation (H3K4me3) were markedly increased at all the tested target promoters upon knockdown of either ZNF516, CtBP1, or CoREST (Fig. 5e). These results indicate that CtBP1 and CoREST are recruited on target gene promoters by ZNF516 in transcription repression of target genes by the ZNF516–CtBP/LSD1/CoREST complex.

**Transcription repression of EGFR by ZNF516-associated complex.** As stated before, overexpression/dysregulation of

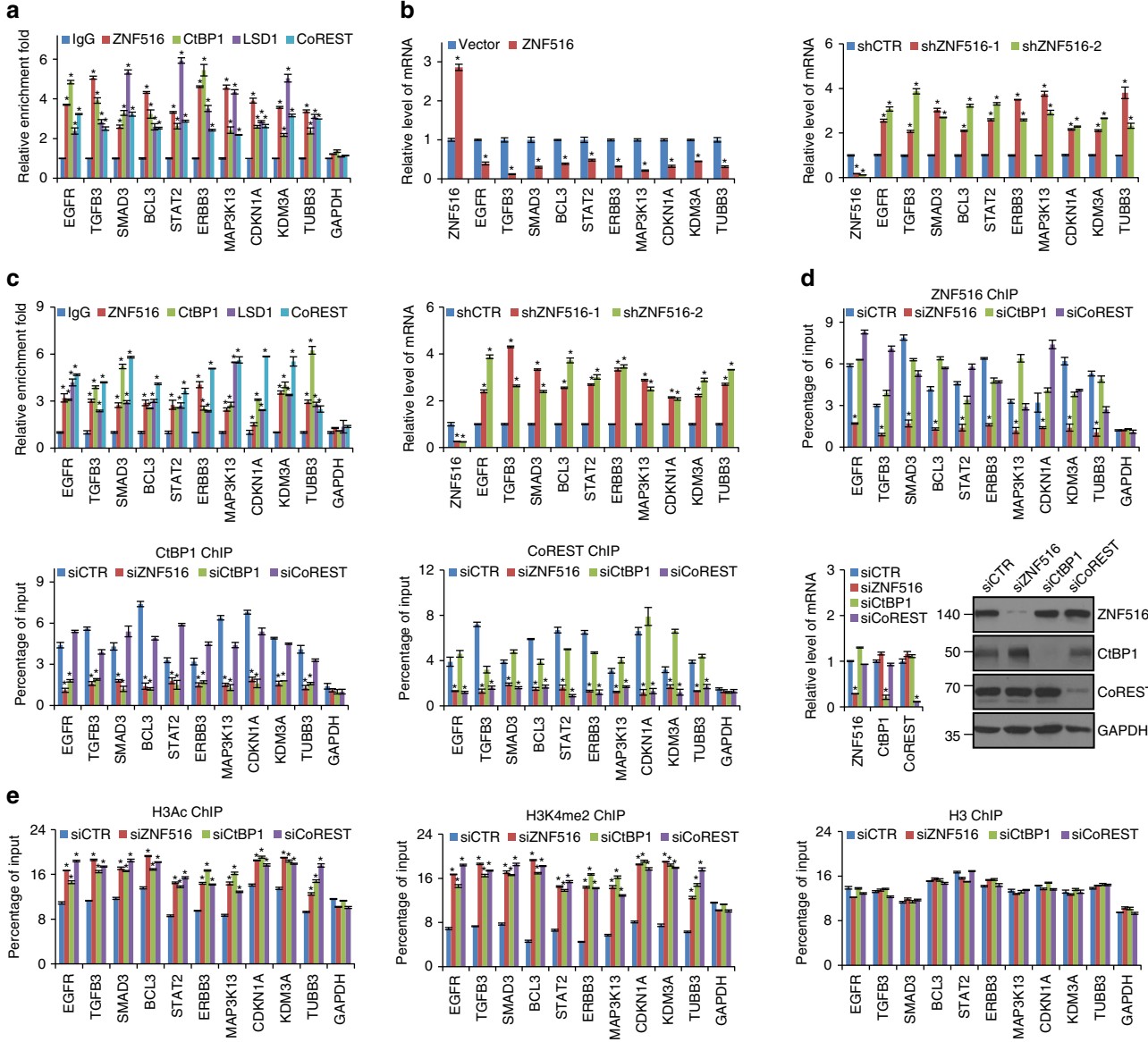

**Fig. 5** Verification of the ChIP-seq results in breast cancer cells. **a** Verification of the ChIP-seq results by qChIP analysis of the indicated genes. MCF-7 cells were collected and qChIP experiments were performed with indicated antibodies. Results are represented as fold change over control IgG with *GAPDH* as a negative control. **b** Total cellular RNAs were prepared from MCF-7 cells with ZNF516 overexpression or knockdown for quantitative real-time RT-PCR analysis of the indicated genes. **c** Verification of the ChIP-seq results by qChIP analysis (*left*) and quantitative real-time RT-PCR analysis (*right*) of the indicated genes in MDA-MB-231 cells. **d** MCF-7 cells were transfected with the control, ZNF516, CtBP1, or CoREST siRNAs. qChIP analysis of the selected promoters was performed using antibodies against indicated proteins. Results are represented as fold change over control IgG with *GAPDH* as a negative control. The knockdown efficiencies of ZNF516, CtBP1, and CoREST were verified by real-time RT-PCR and western blotting. **e** MCF-7 cells were transfected with the control, ZNF516, CtBP1, or CoREST siRNAs. qChIP analysis of the selected promoters was performed using antibodies against H3Ac, or H3K4me2. H3 was detected as an internal control. Each *bar* represents the mean ± S.D. for triplicate experiments. *P*-values were determined by Student's *t*-test. *$P < 0.05$

*EGFR* is a frequent event in the development and progression of tumors from a broad spectrum of tissue origins[3–7]. In addition, genetic study revealed that amplifications in the *EGFR* gene are restricted to regions of the regulatory sequence in the 5′-end of intron 1 and associated with EGFR expression in epithelial breast tumors[14], indicating the importance of transcriptional regulation of EGFR in breast carcinogenesis. Thus, our observation that EGFR is targeted by the ZNF516–CtBP/LSD1/CoREST complex is of great significance. To further confirm the transcription repression of EGFR by the ZNF516–CtBP/LSD1/CoREST complex, ZNF516 was overexpressed or knocked down in MCF-7 or MDA-MB-231 cells, and western blotting analysis showed that gain-of-function of ZNF516 was associated with a reduced

expression of EGFR, whereas loss-of-function of ZNF516 was accompanied by an elevated expression of EGFR (Fig. 6a). Significantly, the reduced expression of EGFR mRNA and protein associated with ZNF516 overexpression was diminished when CtBP1, LSD1, or CoREST was depleted in MCF-7 cells and MDA-MB-231 cells, as measured by real-time RT-PCR and western blotting, respectively (Fig. 6b and Supplementary Fig. 3a).

We next cloned a 543 bp fragment from *EGFR* promoter (Chr 7: 55,085,122–55,085,664), where ZNF516 peak was located in ChIP-seq assays, and constructed a luciferase reporter to test whether ZNF516 is capable of repressing the *EGFR* promoter-driven luciferase reporter (*EGFR*-Luc). In these experiments, MCF-7 cells were co-transfected with *EGFR*-Luc and ZNF516

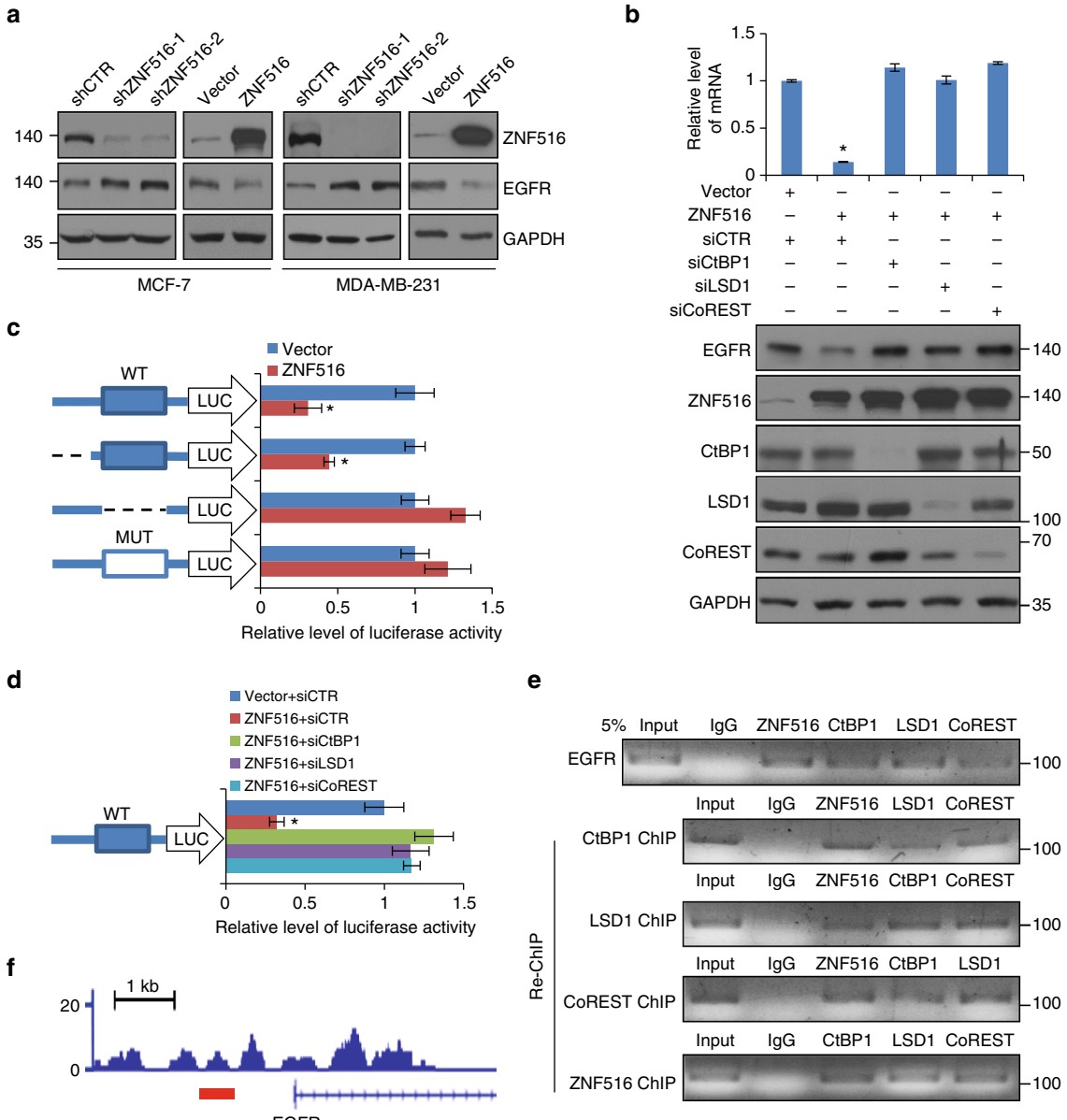

**Fig. 6** Transcription repression of EGFR by ZNF516-CtBP/LSD1/CoREST complex. **a** Total cellular proteins were prepared from MCF-7 and MDA-MB-231 cells with ZNF516 overexpression or knockdown and analyzed for EGFR protein expression by western blotting. **b** MCF-7 cells were transfected with siRNAs of control, CtBP1, CoREST, or LSD1 together with empty vector or ZNF516 expression constructs. The mRNA or protein level of EGFR was measured by real-time RT-PCR or western blotting. **c** MCF-7 cells were co-transfected with *EGFR*-Luc wild-type, deletion, or mutants and expression construct for ZNF516. Forty-eight hours after the transfection, luciferase activity was measured. Relative luciferase activity was calculated as firefly luciferase activity divided by renilla luciferase activity and shown relative to the control. **d** MCF-7 cells were transfected control, CtBP1, LSD1 or CoREST siRNAs together with *EGFR*-Luc construct and ZNF516 expression constructs. Forty-eight hours after the transfection, luciferase activity was measured. Relative luciferase activity was calculated as firefly luciferase activity divided by renilla luciferase activity and shown relative to the control. **e** Co-occupancy of the *EGFR* promoter by ZNF516 and the components of CtBP/LSD1/CoREST complex. Soluble chromatin from MCF-7 cells was prepared for ChIP and Re-ChIP assays with antibodies against the indicated proteins. **f** The ChIP-seq track at *EGFR* promoter. The *red line* indicates the binding site of ZNF516 on the *EGFR* promoter. Each *bar* represents the mean ± S.D. for triplicate experiments. *P*-values were determined by Student's *t*-test. *$P < 0.05$

expression plasmid. Cellular lysates were prepared and luciferase activity was analyzed. As shown in Fig. 6c, ZNF516 was able to repress the *EGFR* promoter activity, but only when CCTCC motif existed; *EGFR* promoter lack of this sequence did not respond to ZNF516. Consistently, ZNF516 was no longer able to repress *EGFR*-Luc activity when CtBP1, LSD1, or CoREST was depleted (Fig. 6d), further supporting the targeting of EGFR gene by the ZNF516–CtBP/LSD1/CoREST complex. Similar results were obtained in MDA-MB-231 cells (Supplementary Fig. 3b, c).

To further support the argument that ZNF516, CtBP1, LSD1, and CoREST form one protein complex on target promoters, ChIP/Re-ChIP experiments were performed on *EGFR* promoter in MCF-7 cells. In these experiments, soluble chromatins were first immuno-precipitated with antibodies against ZNF516, CtBP1, LSD1, or CoREST. The immunoprecipitates were subsequently re-immunoprecipitated with appropriate antibodies. The results showed that, in precipitates, *EGFR* promoter that was immunoprecipitated with antibodies against ZNF516 could be re-immunoprecipitated

with antibodies against CtBP1, LSD1, or CoREST (Fig. 6e). Similar results were obtained when initial ChIP was done with antibodies against CtBP1, LSD1, or CoREST (Fig. 6e). The ChIP-seq track at *EGFR* promoter was shown in Fig. 6f. Collectively, these data support our argument that the ZNF516–CtBP/LSD1/CoREST complex targets and represses the transcription of *EGFR*.

**ZNF516 suppresses the growth and metastasis of breast cancer.** The identification of EGFR as a target of the ZNF516–CtBP/LSD1/CoREST complex and the well-documented role of EGFR in the development and progression of of various malignancies[3–5] suggest that the ZNF516–CtBP/LSD1/CoREST complex may also function in breast cancer growth and metastasis. To investigate

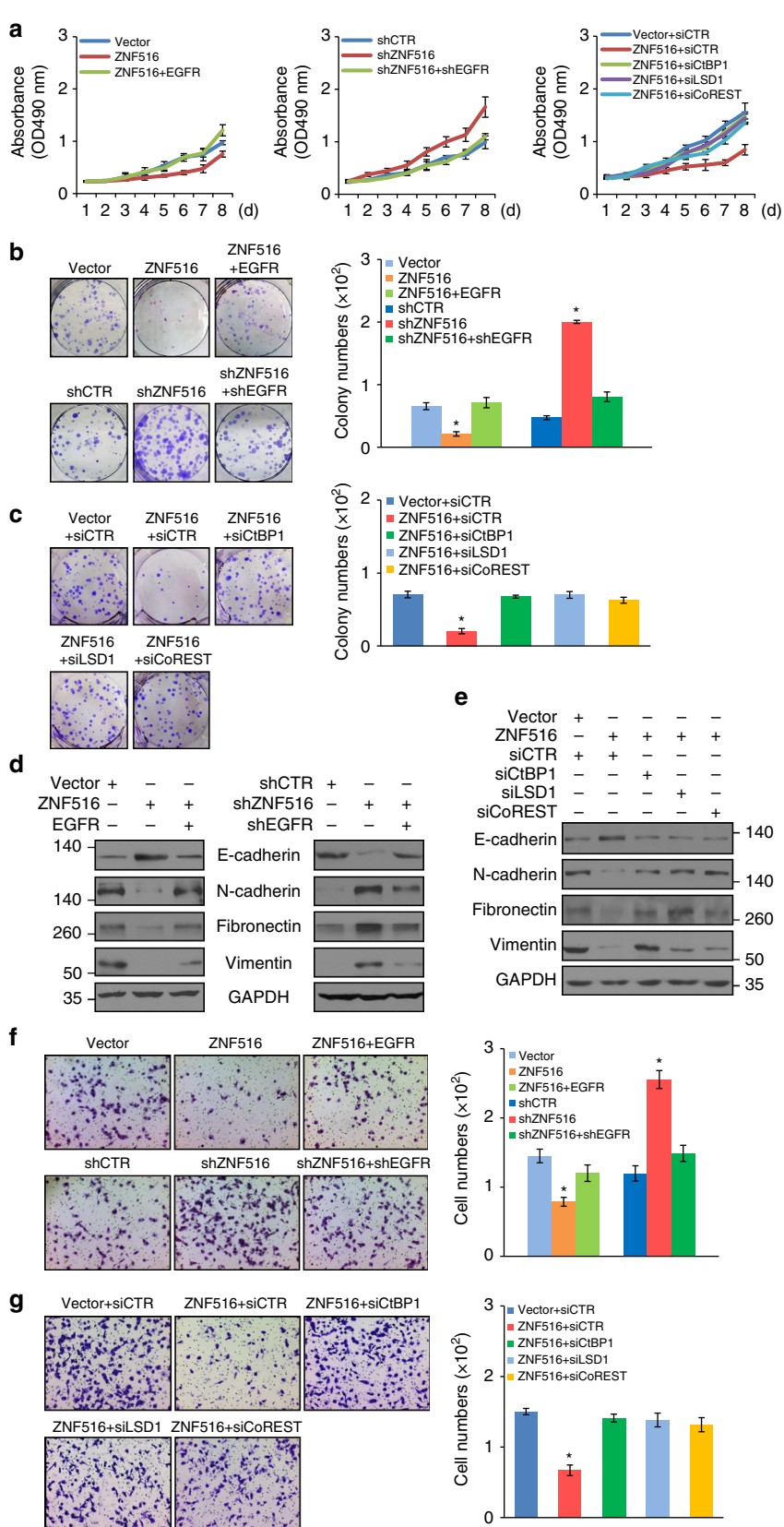

this hypothesis, we first tested the effect of the ZNF516–CtBP/LSD1/CoREST complex on the proliferation of breast cancer cells in vitro. To this end, gain-of-function or loss-of-function of ZNF516 was performed in MCF-7 and MDA-MB-231 cells and the proliferation of these cells was examined by MTS assays. The results showed that ZNF516 overexpression had a significant inhibitory effect on breast cancer cell proliferation, a phenotype that could be rescued by simultaneous overexpression of EGFR (Fig. 7a and Supplementary Fig. 4a, b). Consistently, depletion of ZNF516 promoted breast cancer cell proliferation, an effect that could be abrogated, at least partially, by co-knockdown of EGFR (Fig. 7a). Moreover, the negative effect of ZNF516 overexpression on cell proliferation was probably through the ZNF516–CtBP/LSD1/CoREST complex, as simultaneous depletion of either CtBP1, LSD1, or CoREST attenuated the effect (Fig. 7a and Supplementary Fig. 4b). Moreover, colony formation assays in MCF-7 and MDA-MB-231 cells showed that overexpression of ZNF516 was associated with a decreased colony number, a phenotype that could be, at least partially, rescued by overexpression of EGFR (Fig. 7b). Conversely, knockdown of ZNF516 resulted in an increased in colony number, which was abrogated upon co-knockdown of EGFR (Fig. 7b). Consistently, the effect of ZNF516 overexpression on colony number was offset by simultaneous depletion of CtBP1, LSD1, or CoREST (Fig. 7c and Supplementary Fig. 4c). Altogether, these experiments support a role for ZNF516 in the inhibition of cell proliferation and indicate that ZNF516 does so through association with the CtBP/LSD1/CoREST complex and downregulation of EGFR expression.

Breast cancer metastasis remains a major health problem associated with exceptionally poor patient survival, and many lines of evidence indicate EGFR has a pivotal role in increased cell motility, invasiveness, and progressive regional and distant metastasis of breast cancer[48]. To investigate whether or not the ZNF516–CtBP/LSD1/CoREST complex regulates the invasive potential of breast cancer cells in vitro, we first measured the expression of epithelial/mesenchymal markers by western blotting in MCF-7 and MDA-MB-231 cells, as epithelial–mesenchymal transition (EMT) is a hallmark of cancer and an early event in cell invasion/cancer metastasis[49]. The results showed that overexpression of ZNF516 resulted in induction of epithelial protein markers and reduction of mesenchymal markers (Fig. 7d and Supplementary Fig. 4d), and overexpression of EGFR, at least in part, counteracted the effect of ZNF516 on the expression of epithelial/mesenchymal markers (Fig. 7d). Conversely, depletion of ZNF516 resulted in the reduction of epithelial markers and induction of mesenchymal markers, which were partially attenuated by co-knockdown of EGFR (Fig. 7d). Moreover, overexpression of ZNF516 no longer led to the alteration of the expression of epithelial/mesenchymal markers when CtBP1, LSD1, or CoREST was depleted (Fig. 7e and Supplementary Fig. 4d). Collectively, these results support a role for ZNF516 in the regulation of EMT and indicate that ZNF516 does so through

association with the CtBP/LSD1/CoREST complex and downregulation of EGFR expression.

We then performed transwell cell invasion and tested the effect of ZNF516 on the invasive potential of breast cancer cells. The MDA-MB-231 cells are highly metastatic and are extensively used as a well-established model for invasion and metastasis assays[50]. For this purpose, MDA-MB-231 cells were infected with lentiviruses carrying ZNF516 and/or EGFR or infected with lentiviruses carrying ZNF516 shRNA and/or EGFR shRNA. Transwell cell invasion assays showed that overexpression of ZNF516 was associated with a decrease in the invasive potential and knockdown of ZNF516 was accompanied by an increase in the invasive potential of cells (Fig. 7f). In addition, the inhibitory effect of ZNF516 overexpression on the invasive potential of cells could be, at least partially, offset by overexpression of EGFR, whereas the increase in the invasive potential associated with ZNF516 knockdown could be partially offset by co-knockdown of EGFR (Fig. 7f). Consistently, the effect of by ZNF516 overexpression on the decreased invasive potential was attenuated by simultaneous depletion of CtBP1, LSD1, or CoREST (Fig. 7g). These results support a role for ZNF516 in the regulation of the invasive potential of breast cancer cells and indicate that ZNF516 does so through downregulation of EGFR expression.

To investigate the role of ZNF516 in the growth and metastasis of breast cancer in vivo, MDA-MB-231 cells engineered to stably express firefly luciferase (MDA-MB-231-Luc-D3H2LN) were infected with lentiviruses carrying ZNF516 or/and EGFR, or infected with lentiviruses carrying shZNF516 or/and shEGFR. These cells were then orthotopically implanted onto the abdominal mammary fat pad ($n = 6$) of 6-week-old female immunocompromised severe combined immunodeficiency (SCID) mice. The growth/dissemination of tumors was monitored weekly by bioluminescence imaging with IVIS imaging system. Tumor metastasis was measured by quantitative bioluminescence imaging after 8 weeks. A metastatic event was defined as any detectable luciferase signal above background and away from the primary tumor site. The results showed that overexpression of ZNF516 suppressed the growth of the primary tumor and the metastasis of the MDA-MB-231-Luc-D3H2LN tumors to lung or liver, whereas ZNF516 depletion promoted the growth of the primary tumor and the metastasis of the MDA-MB-231-Luc-D3H2LN tumors to lung, liver, and spleen (Fig. 8a, b). However, the growth of the primary tumor and metastasis were significantly enhanced in mice implanted with MDA-MB-231-Luc-D3H2LN tumors overexpressing ZNF516 + EGFR compared with tumors overexpressing ZNF516 and were significantly inhibited in mice implanted with MDA-MB-231-Luc-D3H2LN tumors carrying shZNF516 + shEGFR compared with tumors carrying shZNF516 (Fig. 8a, b). These results indicate that ZNF516 suppresses the growth and metastasis of breast cancer, and that it does so, at least in part, through repression of EGFR.

**Fig. 7** ZNF516 inhibits the proliferation and invasion of breast cancer cells in vitro. **a** MCF-7 cells were infected with lentiviruses carrying FLAG-tagged ZNF516 or/and EGFR (*left*), or with lentiviruses carrying ZNF516 shRNA or/and EGFR shRNA (*middle*), or MCF-7 cells were transfected with the control, CtBP1, CoREST, or LSD1 siRNAs together with expression constructs for ZNF516 (*right*). Cells were split into 96-well plates and then harvested at indicated day. The growth curves of the cells were measured with MTS assay. Each *point* represents the mean ± S.D. for three independent experiments. **b** MCF-7 cell stably transfected with indicated plasmids were maintained for 14 days before staining with crystal violet and counting for colony numbers in colony formation assay. **c** MCF-7 cells transfected with indicated siRNAs and expression plasmids were tested for colony formation assay. **d** MCF-7 cells were infected with lentiviruses carrying indicated plasmids, and the expressions of epithelial and mesenchymal protein markers were tested by western blotting. **e** The expressions of the epithelial and mesenchymal markers were measured by western blotting in MCF-7 co-transfected with indicated siRNAs and expression plasmids. **f** MDA-MB-231 cells stably transfected with indicated plasmids were starved for 18 h before cell invasion assays were performed using Matrigel transwell filters. The invaded cells were stained and counted. **g** MDA-MB-231 cells transfected with indicated siRNAs and expression plasmids were starved for 18 h before cell invasion assays were performed using Matrigel transwell filters. The invaded cells were stained and counted. Each *bar* represents the mean ± S.D. for triplicate experiments. *P*-values were determined by Student's *t*-test. *$P < 0.05$

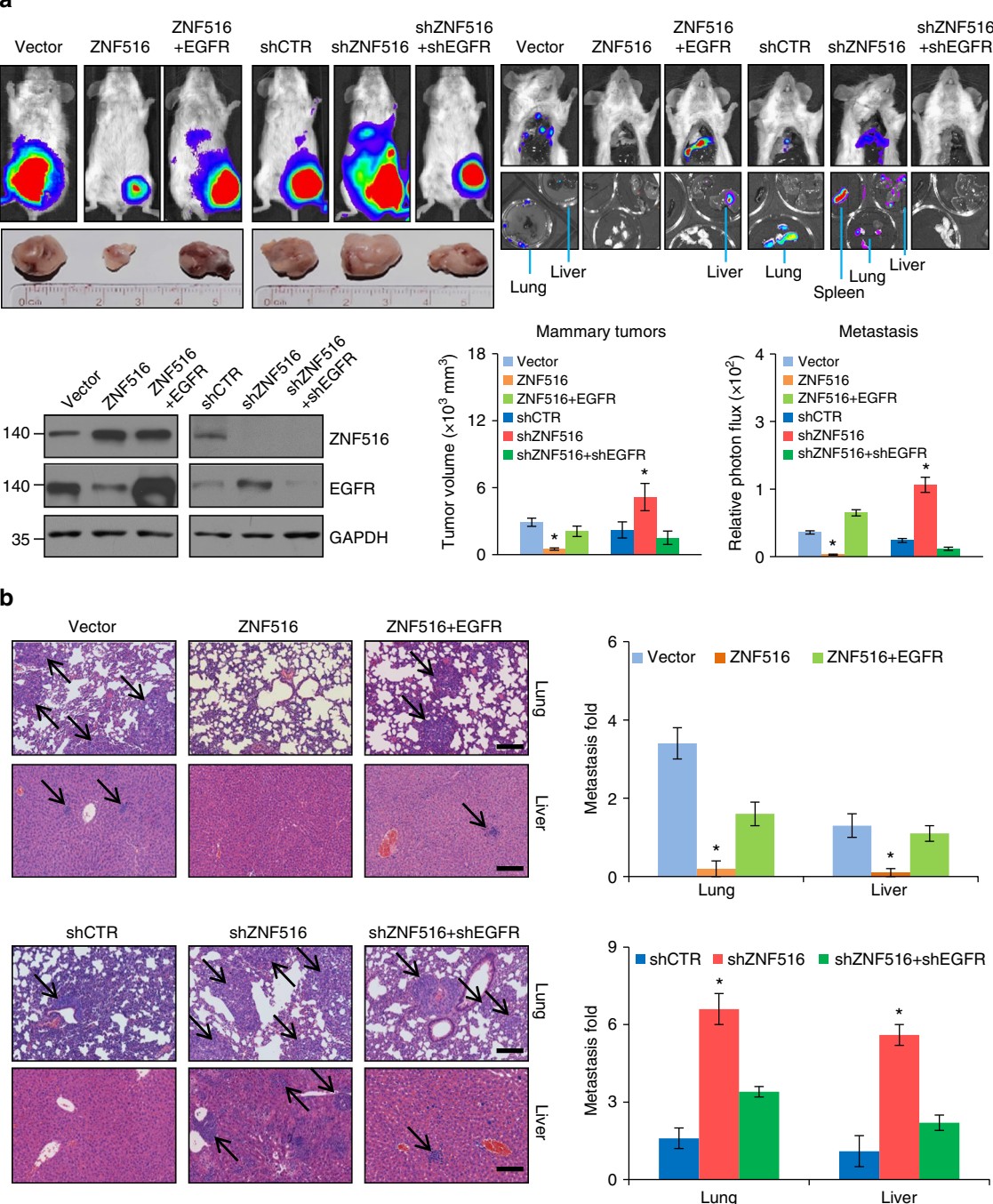

**Fig. 8** ZNF516 suppresses the growth and metastasis of breast cancer in vivo. **a** MDA-MB-231-Luc-D3H2LN cells infected with lentiviruses carrying indicated plasmids were inoculated into the left abdominal mammary fat pad ($5 \times 10^6$ cells) of 6-week-old immunocompromised female SCID beige mice. Tumor size was measured after 8 weeks (mammary tumors, $n = 6$). Lung, liver, or spleen metastases were quantified using bioluminescence imaging after 6 weeks of initial implantation, and representative in vivo bioluminescent images are shown. The efficiency of lentivirus-mediated gene expression or depletion in MDA-MB-231-Luc-D3H2LN cells was verified by western blotting. **b** Representative images of lung or liver sections stained with H&E are shown. *Scale bar* 200 μm. In all *panels*, each *bar* represents the mean ± S.D. ($n = 6$). *P*-values were determined by Student's *t*-test. *$P < 0.05$

**Clinicopathological relevance of ZNF516 in breast carcinomas**. To gain further support of the role of the ZNF516–CtBP/LSD1/CoREST–EGFR axis in the development and progression of breast cancer and to extend our observations to a clinicopathologically relevant context, we first profiled the expression pattern of ZNF516 and EGFR in breast cancer cell lines. Western blotting analysis showed the level of ZNF516 is negative correlated with that of EGFR (Fig. 9a), consistent with our working model. We then collected 20 samples of triple-negative breast cancer paired with adjacent normal mammary tissues and analyzed by western blotting for the expression of ZNF516 and EGFR. We found that the protein levels of ZNF516 were lower in tumor samples than in adjacent tissues, whereas the levels of EGFR showed an inverse trend (Fig. 9b). Quantitation and statistical analysis revealed that the relative level of ZNF516 protein was negatively correlated with that of EGFR (Supplementary Table 1).

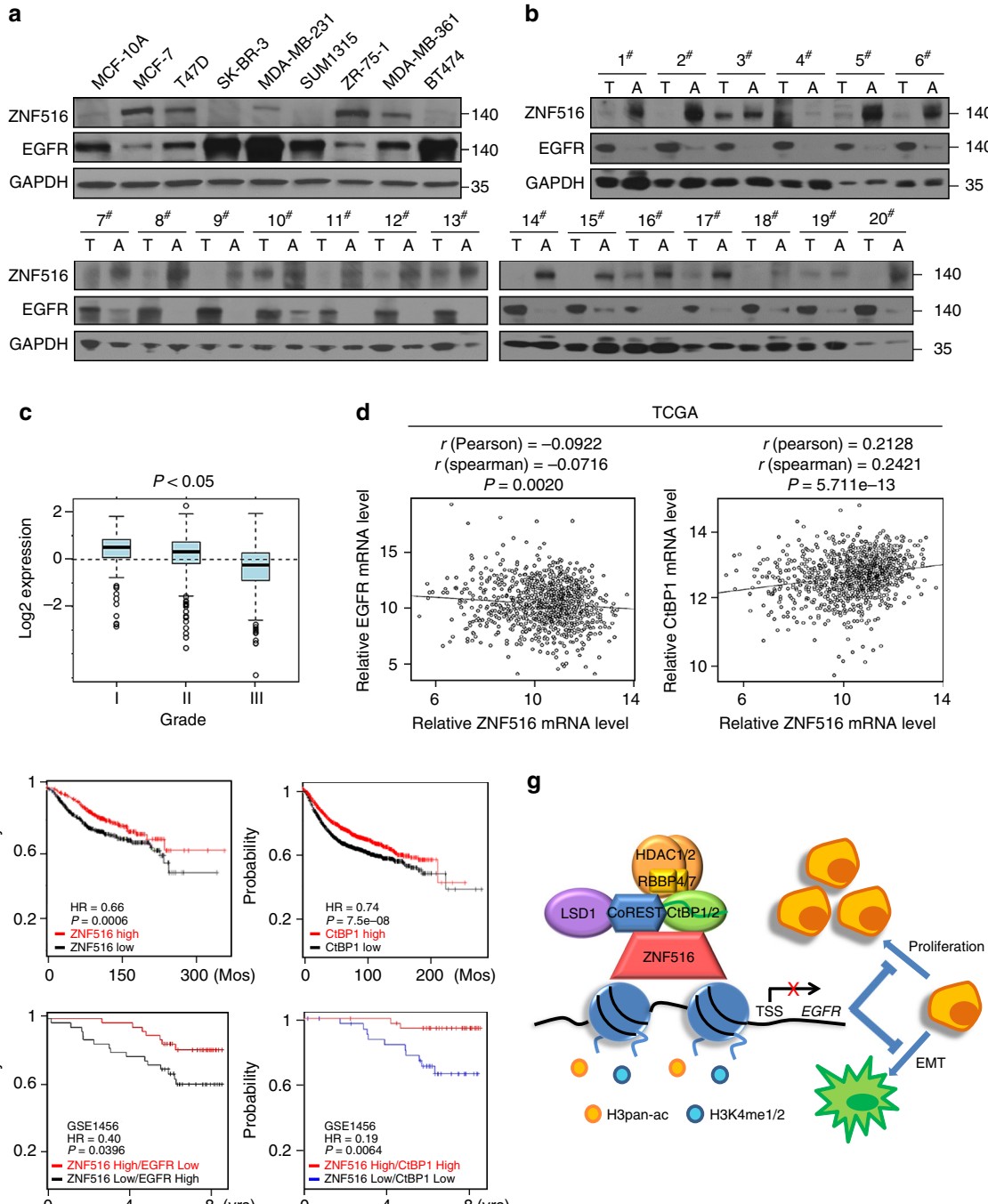

**Fig. 9** Clinicopathological relevance of ZNF516 in breast carcinomas. **a** Western blotting analysis of ZNF516 and EGFR expression in indicated cell lines. **b** Total proteins from 20 paired samples of triple-negative breast cancer (*T*) versus adjacent normal breast tissues (*A*) were extracted for western blotting analysis with antibodies against ZNF516 or EGFR. **c** The correlation of ZNF516 expression and histological grade using the Gene expression-based Outcome for Breast cancer Online (GOBO) analysis (http://co.bmc.lu.se/gobo/gsa.pl). The number of tumor samples in each grade is grade I (*n* = 239), grade II (*n* = 677), and grade III (*n* = 495). *P*-value is calculated using one-way ANOVA. **d** Analysis of TCGA data set for the correlations in mRNA expression between ZNF516 and EGFR, or CtBP1. The relative level of EGFR or CtBP1 was plotted against that of ZNF516. **e** Kaplan–Meier survival analysis for the relationship between survival time and ZNF516 or CtBP1 signature in breast cancer using the online tool (http://kmplot.com/analysis/). **f** Kaplan–Meier survival analysis of the published data sets (GSE1456) for the relationship between survival time and ZNF516/EGFR or ZNF516/CtBP1 signature in breast cancer. **g** Proposed model of the ZNF516-CtBP/LSD1/CoREST complex in repressing *EGFR* oncogene to inhibit proliferation and invasion of breast cancer cells

Interrogation of public dataset for ZNF516 expression using the Gene expression-based Outcome for Breast cancer Online (GOBO) tool (http://co.bmc.lu.se/gobo/gsa.pl) support the notion that the level of ZNF516 expression is negatively correlated with the histological staging of the tumors (Fig. 9c). Moreover, the

expression of ZNF516 is higher in luminal A and luminal B subtypes and lower in HER2-enriched and basal-like subtypes, whereas the levels of EGFR showed an inverse trend (Supplementary Fig. 5a). In addition, using TCGA database, we found a negative correlation in mRNA expression of ZNF516 and EGFR,

as well as a positive correlation of ZNF516 and CtBP1 (Fig. 9d). Similarly, querying published clinical data sets (GSE42568, GSE36774, and GSE21653) showed a clear negative correlation of mRNA levels between ZNF516 and EGFR (Supplementary Fig. 5b). Furthermore, analysis of published clinical data sets (GSE21653) revealed negative correlations of mRNA levels between ZNF516 and EGFR in different subtypes of breast cancers (Supplementary Fig. 5c). Finally, to further extend our observations to a clinicopathologically relevant setting, we performed Kaplan–Meier survival analysis (http://kmplot.com/analysis/) of public data sets and found that higher ZNF516 or CtBP1 expression is associated with a better relapse-free survival of breast cancer patients, when the influence of systemic treatment, endocrine therapy, and chemotherapy were excluded (Fig. 9e). Moreover, higher ZNF516 expression is associated with a better relapse-free survival of breast cancer patients of luminal A and even basal-like subtypes (Supplementary Fig. 5d). Further stratification of patient groups based on the inverse expression of ZNF516 and EGFR or the coexpression of ZNF516 and CtBP1 improved the predictive capability of ZNF516 (GSE1456) (Fig. 9f). Collectively, these analyses support our observations that ZNF516 is a transcription repressor and a potent suppressor of breast development and progression.

## Discussion

*EGFR* dysregulation/EGFR overexpression is a frequent event in malignancies from a broad spectrum of tissue origins[3–7] and this aberration is believed to be associated with a more aggressive phenotype and accordingly worse survival of the cancer patients[8]. Intriguingly, it has been documented that amplifications in the *EGFR* gene are restricted to regions of the regulatory sequence in the 5′-end of intron 1 and associated with EGFR expression in epithelial breast tumors[14], suggesting that transcriptional regulation of *EGFR* might be pathologically relevant in breast cancer carcinogenesis by contributing to EGFR overexpression. Therefore, understanding of the regulation of EGFR expression is of great importance for cancer prevention and intervention.

The 5′-regulatory sequence of *EGFR* is somewhat unique in that it contains a GC-rich promoter without any consensus sequences, such as TATA or CAAT boxes[14]. It has been reported that basal transcription of *EGFR* is regulated by the transcription factor Sp1[51]. It has been shown that *EGFR* is transactivated by p53[52], retinoic acid receptor γ (RAR-γ)[53], and Stat5b[54], and transrepressed by EGFR transcriptional repressor (ETR)[55], GC-binding factor (GCF)[56], TGF-β inducible early gene 1 (TIEG1)[57], and receptor-interacting protein (RIP1)[58]. We report in the current study that *EGFR* is transrepressed by ZNF516. We showed that ZNF516 does so through recognizing a CCTCC motif in the promoter of *EGFR*. The relative contribution of the above-mentioned transcriptional regulators to the expression of EGFR is probably hard to be determined, and whether different regulators represent differential regulations of EGFR expression in different tissues or under different cellular micromilieu is currently unclear and how different regulators might be coordinated in the regulation of EGFR expression in a specific cell lineage is currently unknown. Nevertheless, in addition to the genetic alterations that lead to *EGFR* amplification, epigenetic-driven EGFR overexpression and its contribution to tumorigenesis have been well-documented and recognized. It is a distinct possibility that the epigenetically-associated aberrations of EGFR expression are imparted by the abnormalities, genetic or/and epigenetic, of transcriptional regulators themselves, as discussed below.

ZNF516 is a member of the Krüppel C2H2-type zinc-finger family proteins[15]. This protein is essential for animal development, as Znf516 null mice die immediately after birth[19]. Previously, ZNF516 has been implicated in both transcription activation[19, 20] and repression[21–24]. In the current study, we showed that ZNF516 is a transcription repressor. We demonstrated that the transcriptional repressive activity of ZNF516 is associated with histone deacetyalse and demethylase activities. Indeed, biochemical studies demonstrated that ZNF516 is associated with the CtBP/LSD1/CoREST complex, which contains both histone deacetyalse and demethylase activities. Significantly, *ZNF516* has been shown to experience frequent copy number loss in colorectal cancer that is associated with chromosomal instability and aneuploidy onset at adenoma–carcinoma transition[25], and hypermethylation of *ZNF516* promoter has been detected in cervical neoplasia that is suggested as a better biomarker[26], underscoring the importance of this transcriptional regulator itself in tumorigenesis. In the current study, we found that the expression of ZNF516 is progressively lost during breast cancer progression, and, consistent with our observation that ZNF516 transcriptionally represses *EGFR*, we showed that the level of ZNF516 expression is negatively correlated with that of EGFR. It is currently unknown if similar genetic or/and epigenetic abnormalities of *ZNF516* also occur in breast cancer, and it is unclear the sequence of the events associated with the aberrant expression of ZNF516 and EGFR. Nevertheless, it is a plausible scenario that loss of ZNF516 expression would lead to the up-regulation of EGFR during breast cancer carcinogenesis.

The CtBP/LSD1/CoREST corepressor complex is recruited by a panel of transcription repressors including ZEB1/2 and ZNF217[36, 37]. It is believed that the recruitment of this corepressor complex depends on the existence of a classical PXDLS and/or RRT motif in a particular transcription repressor[15, 34, 35]. This notion is supported by our current study and both PXDLS and RRT motifs are found in ZNF516. CtBP is a corepressor protein lacking of the ability to bind DNA. Its actions depend on its recruitment to chromatin by various transcription factors capable of DNA binding. Thus, recruitment of CtBP, via its binding clefts, by different transcription factors could elicit different cellular outputs, a possible explanation of the discrepancy of the effects of CtBP on EMT and survival observed in our experiments versus that reported by others[46]. We showed that ZNF516 interacts with the CtBP/LSD1/CoREST corepressor complex and targets the transcription of a cohort of genes and regulates several cellular key biological processes including cell proliferation, apoptosis, invasion, migration and protein catabolic processes. Clearly, the multitude of the cellular function of the ZNF516 is beyond what we investigated in the current study. However, we are by no means to exclude the involvement and the importance of other downstream target genes, besides EGFR. Moreover, it remains to be investigated the scope of and mechanistic insights into the role of ZNF516 in malignancies in other tissue origins. Nevertheless, our results indicate that ZNF516 is a transcription repressor and a potential suppressor of EGFR, adding to the understanding of EGFR-related breast carcinogenesis and supporting the pursuit of ZNF516 as a potential therapeutic target for breast cancer.

## Methods

**Bioinformatics**. The open reading frame, conserved domains, and chromosomal location of *ZNF516* were analyzed using the NCBI databases (www.ncbi.nlm.nih.gov). The theoretical molecular weight of ZNF516 was calculated using www.expacy.ch/tools. The homologous alignment was analyzed by the ClustalW program (version 1.60)[59] and phylogenetic analysis was performed using the Jotun Hein method[60].

**Reagents**. The cDNA for wild-type of ZNF516 was amplified by PCR and ligated into *Bam*HI/*Eco*RI sites of a pcDNA3.1 vector that contains one tag of FLAG. The GST-ZNF516-N, GST-ZNF516-ΔC, or GST-ZNF516-C expression plasmid was

constructed by cloning deletion mutants of ZNF516 into a pGEX-4T-3 vector. All clones were confirmed by DNA sequencing. pCS2-HA-CoREST was from Dr. Hongtao Yu (The University of Texas Southwestern Medical Center, USA). pCAG-wtEGFR-IRES-Neomycin was from Dr. Liang Chen (National Institute of Biological Sciences, China). ZNF516, CtBP1, CtBP2, or CoREST deletion mutants were prepared by separate fragments of full length, and point mutants of EGFR promoter was generated with Stratagene mutagenesis kit. Antibodies used: anti-ZNF516 (Bethyl, A303-392A, IP, 1:300 for IF and 1:1000 for WB); anti-FLAG (Sigma, F3165, IP and 1:10,000 for WB); anti-fibronectin (Sigma, F7387, 1:800 for WB); anti-GAPDH (MBL, M171-3, 1:3000 for WB); anti-E-cadherin (Cell Signaling Technology, #3195, 1:1000 for WB); anti-N-cadherin (Cell Signaling Technology, #13116, 1:1000 for WB); anti-vimentin (Cell Signaling Technology, #5741, 1:1000 for WB); anti-CoREST (Santa Cruz, sc-376567, IP and 1:500 for WB); anti-CtBP1 (Proteintech, 10972-1-AP, IP and 1:1000 for WB)/(Abgent, AT1665a, 1:500 for WB); anti-CtBP2 (ABclonal Technology, A2257, IP and 1:1000 for WB)/(Origene, TA807909, 1:1000 for WB); anti-LSD1 (Abcam, ab17721, IP and 1:2000 for WB)/(Abcam, ab17721, IP and 1:2000 for WB); ani-HDAC1 (Abcam, ab7028, IP and 1:2000 for WB)/(Proteintech, 66085-1-Ig, 1:2000 for WB); ani-HDAC2 (Abcam, ab7029, IP and 1:2000 for WB)/(ABclonal Technology, A2084, 1:500 for WB); anti-EGFR (Ruiying Biological, RLT1485, 1:1000 for WB). Anti-FLAG M2 affinity gel (A2220), FLAG peptide (F3290), and trichostatin A (T1952) were purchased from Sigma. Tranylcypromine (X042-1EA) was purchased from Arborassays.

**Cell culture and transfection.** The cell lines used were obtained from the American Type Culture Collection (ATCC). Cells were maintained according to the ATCC's recommendation. MCF-7 and HEK293T cells were maintained in Dulbecco's modified Eagle's medium (DMEM) supplemented with 10% fetal bovine serum (FBS). Cells were maintained in a humidified incubator equilibrated with 5% $CO_2$ at 37 °C. MDA-MB-231 cells were cultured in L-15 medium supplemented with 10% FBS and without $CO_2$. All of the cells were authenticated by examination of morphology and growth characteristics, and were confirmed to be mycoplasma-free using the Mycoplasma Detection Kit (Biotool). None of the cell lines used for this study are listed in the database of commonly misidentified cell lines maintained by ICLAC. The sequences of siRNA were: control siRNA, 5′-UUCUCCGAACGUGUCACGU-3′; CtBP siRNA-1, 5′-GGGAGGACCUGGA-GAAGUU-3′; CtBP siRNA-2, 5′-ACGACUUCACCGUCAAGCA-3′; LSD1 siRNA-1, 5′-AAGGAAAGCUAGAAGAAAA-3′; LSD1 siRNA-2, 5′-ACA-CAAGGAAAGCUAGAAGAA-3′; CoREST siRNA-1, 5′-CAAAGUUGGAU-GAAUACAU-3′; CoREST siRNA-2, 5′-CCAGAUAAAAUCUAUAGCAA-3′. All of the siRNAs were synthesized by GeneChem Inc. (Shanghai, China). The siRNA oligonucleotides were transfected into cells using RNAiMAX (Invitrogen) with the final concentration at 20 nM. Transfections of expression plasmids in HEK293T or MCF-7 cells were carried out using poly(ethylene imine) (PEI) (Polysciences) according to the manufacturer's recommendations. Transfections of expression plasmids in MDA-MB-231 cells were carried out using Lipofectamine 2000 (Invitrogen) as the recommendations.

**Lentiviral production and infection.** The generation of the PITA-ZNF516, PITA-EGFR, pLKO.1-shZNF516, or pLKO.1-shEGFR lentiviruses was conducted according to a protocol described by Addgene (http://www.addgene.org/tools/protocols/plko/#E). Briefly, the human expression plasmids of PITA-ZNF516 and PITA-EGFR were generated by subcloning the ZNF516 or EGFR fragment into the PITA vector. pLKO.1-shZNF516, and pLKO.1-shEGFR were generated by subcloning shRNA (TRCN0000239483, shZNF516-1; TRCN0000239481, shZNF516-2; TRCN0000039635, shEGFR) into the pLKO.1 vector. The lentiviral plasmid vector, PITA, PITA-ZNF516, PITA-EGFR, pLKO.1, pLKO.1-shZNF516, or pLKO.1-shEGFR, together with psPAX2 and pMD2.G were co-transfected into the packaging cell line HEK293T. Viral supernatants were collected 48 h later, clarified by filtration, and concentrated by ultracentrifugation. The concentrated virus was used to infect $5 \times 10^5$ cells (20–30% confluent) in a 60 mm dish with 5 µg/ml polybrene. Infected cells were selected by 2 µg/ml puromycin and/or neomycin (Merck). For the re-expressing and re-silencing EGFR experiments, the level of EGFR expression was controlled by creating stable clones of cells that were expressing different levels of EGFR, and the clones with EGFR levels close to original EGFR level were chosen for phenotype experiments.

**Luciferase reporter assay.** HEK293T, MCF-7, or MDA-MB-231 cells in 24-well plates were transfected with luciferase reporter, renilla, and indicated expression constructs. The amount of DNA in each transfection was kept constant by addition of empty vector. Thirty-six hours after transfection, the firefly and renilla luciferase were assayed according to the manufacturer's protocol (Promega), and the firefly luciferase activity was normalized to that of renilla luciferase. Each experiment was performed in triplicate and repeated at last three times.

**Fluorescence confocal microscopy.** MCF-7 or MDA-MB-231 cells growing on six-well chamber slides were washed with PBS, fixed in 4% (w/v) paraformaldehyde, permeabilized with 0.1% (v/v) Triton X-100 in PBS, blocked with 0.8% BSA, and incubated with appropriate primary antibodies followed by staining with FITC-conjugated secondary antibodies (Jackson ImmunoResearch). The cells were

washed four times and a final concentration of 0.1 µg/ml DAPI (Sigma) was included in the final washing to stain nuclei. Images were visualized with an Olympus inverted microscope equipped with a charge-couple camera.

**Immunopurification and mass spectrometry.** HEK293T cells stably expressing FLAG-ZNF516 were washed twice with cold PBS, scraped, and collected by centrifugation at 1500×g for 5 min. Cellular extracts were prepared by incubating the cells in lysis buffer containing protease inhibitor cocktail (Roche). Anti-FLAG immunoaffinity columns were prepared using anti-FLAG M2 affinity gel (Sigma) following the manufacturer's suggestions. Cell lysates were obtained from about $5 \times 10^8$ cells and applied to an equilibrated FLAG column of 1-ml bed volume to allow for adsorption of the protein complex to the column resin. After binding, the column was washed with cold PBS plus 0.1% Nonidet P-40. FLAG peptide (Sigma) was applied to the column to elute the FLAG protein complex as described by the vendor. Fractions of the bed volume were collected and resolved on NuPAGE 4-12% Bis-Tris gel (Invitrogen), silver-stained using Pierce silver stain kit, and subjected to LC-MS/MS (Agilent 6340) sequencing.

**Fast protein liquid chromatography.** Cellular lysates of HEK293T cells stably expressing FLAG-ZNF516 were prepared by incubating the cells in lysis buffer containing protease inhibitor cocktail (Roche). Anti-FLAG immunoaffinity columns were prepared using anti-FLAG M2 affinity gel (Sigma) following the manufacturer's suggestions. Cell lysates were obtained and applied to an equilibrated FLAG column of 1-ml bed volume to allow for adsorption of the protein complex to the column resin. After binding, the column was washed with cold PBS plus 0.1% Nonidet P-40. FLAG peptide (Sigma) was applied to the column to elute the FLAG protein complex as described by the vendor. Fractions of the bed volume were collected and concentrated to 0.5 ml using a Millipore Ultrafree centrifugal filter apparatus (3 kDa nominal molecular mass limit), and then applied to an $850 \times 20$ mm Superose 6 size exclusion column (Amersham Biosciences) that was equilibrated with PBS and calibrated with protein standards (blue dextran, 2000 kDa; thyroglobulin, 669 kDa; ferritin, 440 kDa; aldolase, 158 kDa; ovalbumin, 43 kDa; all from Amersham Biosciences). The column was eluted at a flow rate of 0.5 ml/min and fractions were separately collected. MCF-7 cell nuclear extracts were prepared and dialyzed against buffer D (20 mM HEPES [pH 8.0], 10% glycerol, 0.1 mM EDTA, 300 mM NaCl) (Applygen Technologies Inc.). Approximately 5 mg nuclear protein was concentrated to 0.5 ml using a Millipore Ultrafree centrifugal filter apparatus (3 kDa nominal molecular mass limit). Then, the following procedure is the same as above.

**Co-immunoprecipitation and western blotting.** Briefly, cellular lysates were prepared by incubating the cells in lysis buffer (50 mM Tris-HCl, pH 7.5, 150 mM NaCl, 0.5% NP-40, 2 mM EDTA) containing protease inhibitor cocktail (Roche) for 20 min at 4 °C, followed by centrifugation at 14,000×g for 15 min at 4 °C. The protein concentration of the lysates was determined using the BCA protein assay kit (Pierce) according to the manufacturer's protocol. Overall, 5% (1:20) cellular extracts were used for input. For immunoprecipitation, 500 µg of protein was incubated with 2 µg specific antibodies for 12 h at 4 °C with constant rotation; 60 µl of 50% protein A or G agarose beads was then added and the incubation was continued for an additional 2 h. Beads were then washed five times using the lysis buffer. Between washes, the beads were collected by centrifugation at 500×g for 5 min at 4 °C. The precipitated proteins were eluted from the beads by resuspending the beads in 2 × SDS-PAGE loading buffer and boiling for 10 min. The resultant materials from immunoprecipitation or cell lysates were resolved using 10% SDS-PAGE gels and transferred onto nitrocellulose membranes[61]. For western blotting, membranes were incubated with appropriate antibodies for 1 h at room temperature or overnight at 4 °C followed by incubation with a secondary antibody. Immunoreactive bands were visualized using western blotting Luminol reagent (Santa Cruz) according to the manufacturer's recommendation. Uncropped scans with molecular weight reference are shown in Supplementary Fig. 5.

**GST pull-down assay.** GST fusion constructs were expressed in BL21 E. coli bacteria, and crude bacterial lysates were prepared by sonication in TEDGN (50 mM Tris-HCl, pH 7.4, 1.5 mM EDTA, 1 mM dithiothreitol, 10% (v/v) glycerol, 0.4 M NaCl) in the presence of the protease inhibitor mixture. In vitro transcription and translation experiments were done with rabbit reticulocyte lysate (TNT systems, Promega) according to the manufacturer's recommendation. Briefly, equal amounts of GST fusion proteins were immobilized on 50 µl of 50% glutathione-Sepharose 4B slurry beads (Amersham Biosciences) in 0.5 ml of GST pull-down binding buffer (10 mM HEPES, pH 7.6, 3 mM $MgCl_2$, 100 mM KCl, 5 mM EDTA, 5% glycerol, 0.5% CA630). After incubation for 1 h at 4 °C with rotation, beads were washed three times with GST pull-down binding buffer and resuspended in 0.5 ml of GST pull-down binding buffer before adding 10 µl of in vitro transcribed/translated proteins for 2 h at 4 °C with rotation. The beads were then washed three times with binding buffer. The bound proteins were eluted by boiling in 30 µl of 2 × sample loading buffer and resolved on SDS-PAGE[61].

**RT-PCR and real-time RT-PCR.** Total cellular RNAs were isolated from samples with the Trizol reagent (Invitrogen). First strand cDNA synthesis with the Reverse

Transcription System (TransGen Biotech). Quantitation of all gene transcripts was done by qPCR using Power SYBR Green PCR Master Mix and an ABI PRISM 7500 sequence detection system (Applied Biosystems, Foster City, CA) with the expression of *GAPDH* as the internal control. The primer pairs used were: *ZNF516*, 5′-GCACACTCAGTGGTGTTTGAG-3′ (forward) and 5′-GGA-CATCGTGAGGGTACTGC-3′ (reverse); *EGFR*, 5′-CAGGAGGTGGCTGGT-TATG-3′ (forward) and 5′-GCACAGGGCAGGGTTGTT-3′ (reverse); *TGFB3*, 5′-ATGACCCACGTCCCCTATCA-3′ (forward) and 5′-GGCCGAAGGATCTG-GAAGAG-3′ (reverse); *SMAD3*, 5′-GAGTTGAGGCGAAGTTTGGG-3′ (forward) and 5′-ATCCAGGGACCTGGGGATG-3′ (reverse); *BCL3*, 5′-CGTGAACGCG-CAAATGTACT-3′ (forward) and 5′-GATGTCGATGACCCTGCGG-3′ (reverse); *STAT2*, 5′-CGGGACATTCAGGATCCTACC-3′ (forward) and 5′-TCTGATGGGGGTCCAGAGAG-3′ (reverse); *ERBB3*, 5′-TCCTTCCTGCAGT-GGATTCG-3′ (forward) and 5′-CATCTCGGTCCCTCACGATG-3′ (reverse); *MAP3K13*, 5′-CTCTGGGAGAGGGGTGTTTG-3′ (forward) and 5′-CCGTGCCAGCAAATGACATC-3′ (reverse); *CDKN1A*, 5′-GTCACCA-GACTTCTCTGAGCC-3′ (forward) and 5′-ATGGCGCCTGAACAGAAGAAA-3′ (reverse); *KDM3A*, 5′-TGAGCCACACAGACAGGTTG-3′ (forward) and 5′-GCCTGTTTGAACAAGGGCAG-3′ (reverse); *TUBB3*, 5′-AACCA-GATCGGGGCCAAGTT-3′ (forward) and 5′-AGGCACGTACTTGTGAGAA-GAG-3′ (reverse); *GAPDH*, 5′-TCCTCCTGTTTCATCCAAGC-3′ (forward) and 5′-TAGTAGCCGGGCCCTACTTT-3′ (reverse); *CtBP1*, 5′-CCTA-TAGGTACCCTCCGGGC-3′ (forward) and 5′-CTACAACTGGTCACTGGCGT-3′ (reverse); *CoREST*, 5′-ATCGACGCCGCTTCAACATA-3′ (forward) and 5′-CATCCAGAACAGGAGCCTCG-3′ (reverse).

**ChIP sequencing**. MCF-7 cells were maintained in DMEM supplemented with 10% FBS. Approximately $5 \times 10^7$ cells were used for each ChIP-seq assay. The chromatin DNA was precipitated by polyclonal antibodies against ZNF516. The DNA was purified with the Qiagen PCR purification kit. In-depth whole-genome DNA sequencing was performed by the CapitalBio Corporation, Beijing. ChIP-seq samples were typically subjected to strict quality control by the sequencing company, and the results were obtained from a single sample. The raw sequencing data were firstly conducted quality control via Fastx toolkit for the reliable of the following analysis with the default parameters, i.e., only retain sequences with at least 90% base pair with quality score >20. Remained sequences were aligned to the unmasked human reference genome (GRGH37, hg19) using Bowtie Version 2[62] with only one mismatch allowed. MACS Version 2 (Model-based Analysis for ChIP-Seq)[43] was used for the identification of ZNF516-specific binding peaks with all default settings except $q < 0.05$. Genomic distribution of ZNF516 binding sites was analyzed by ChIPseeker[63] with the hg19 genomic annotation, and 3000 bp of upstream and downstream of transcription start sites were considered as promoter regions. De novo motif screening was performed on sequences ±400 bp from the centers of ZNF516 binding peaks using the MEME systems[64]. The motifs within peaks on genes were searched through Find Motif Occurrence (FIMO) scanner[47]. Biological process ontologies analysis was conducted based on the Database for Annotation, Visualization and Integrated Discovery (DAVID, https://david.ncifcrf.gov/).

**ChIP and Re-ChIP**. DNA was purified with the QIAquick PCR Purification Kit. qChIPs were performed using the TransStart Top Green qPCR supermix (Trans-Gen Biotech). Re-ChIPs were done essentially the same as primary IPs. Bead eluates from the first immunoprecipitation were incubated with 10 mM DTT at 37 °C for 30 min and diluted 1:50 in dilution buffer (1% Triton X-100, 2 mM EDTA, 150 mM NaCl, 20 mM Tris-HCl, pH 8.1) followed by re-immunoprecipitation with the second antibodies. The final elution step was performed using 1% SDS solution in Tris-EDTA buffer, pH 8[61, 65]. The sequences of the primers used were: *EGFR*, 5′-GTAGAGCCCGTTCCGTTGTC-3′ (forward) and 5′-AAGCACCACCCATGTGCTTTA-3′ (reverse); *TGFB3*, 5′-GCTCCTTTTCCTGTCCTCCC-3′ (forward) and 5′-ACCCCCTGCA-TACTTCTCCT-3′ (reverse); *SMAD3*, 5′-CCCCTACTTTAGGCTTGGGC-3′ (forward) and 5′-GGTTAGACTGGACAGGGTGC-3′ (reverse); *BCL3*, 5′-GACG-GAGACACGTGAGTGAC-3′ (forward) and 5′-TGTTATGGACGGCTG GTCTTC-3′ reverse); *STAT2*, 5′-ACAGTGGCAGGGTTGGAAAA-3′ (forward) and 5′-GTAGAAACCCAGCCCTGCAT-3′ (reverse); *ERBB3*, 5′-CATCTCTGGG CTTCCGATCC-3′ (forward) and 5′-GCGGCCGATTATGCTTGATG-3′ (reverse); *MAP3K13*, 5′-TGCATTTATAAAAGTACAGCACCCA-3′ (forward) and 5′-ACTCAAGACTGAACAGAATACGC-3′ (reverse); *CDKN1A*, 5′-CAAGGGGGTCTGCTACTGTG-3′ (forward) and 5′-GGGGAGGACAGGC TTCTTTC-3′ (reverse); *KDM3A*, 5′-GGCCCCTACTTTTCCAGTCC-3′ (forward) and 5′-CTCCCCTGTCACAACAGTCC-3′ (reverse); *TUBB3*, 5′-GGTGCGG GTTGGTCTCTAAA-3′ (forward) and 5′-GAGGACAATGGCCCCTCTTG-3′ (reverse).

**Cell invasion assay**. The transwell invasion assay was performed using the transwell chamber (BD biosciences) with a Matrigel-coated filter. Stably infected MDA-MB-231 cells were cultured in Leibovitz's L-15 medium with 10% FBS at 37 °C without $CO_2$. Cells were deprived in serum-free Leibovitz's L-15 medium for 18 h, and harvested. These cells were washed three times in PBS and resuspended in serum-free culture medium. Afterwards, $1 \times 10^5$ of these cells in 0.3 ml of serum-free media were plated onto the upper chamber of the transwell. The upper chamber was then transferred into a well containing 0.5 ml of media supplemented with 10% FBS and incubated for 5 h. Cells may actively migrate from the upper to the lower side of the filter due to FBS as attractant. Cells on the upside were removed using cotton swabs, and the invasive cells on the lower side were fixed, stained with 0.1% crystal violet solution, and counted using light microscope. Each experiment was performed in triplicate and repeated at last three times.

**Cell viability/proliferation assay**. For cell proliferation assays, MCF-7 or MDA-MB-231 cells were seeded into 96-well plates with an equal volume of medium. On the day of harvest, the CellTiter 96® AQueous ONE Solution Reagent (Promega) was added according to the manufacturer's protocol. Plates were incubated at 37 °C for 1 h and cell viability was determined by measuring the absorbance of converted dye at wavelengths 490 nm. The detailed protocol is following the manufacturer's instruction (Promega). Each experiment was performed in triplicate and repeated at last three times.

**Colony formation assay**. MCF-7 or MDA-MB-231 cells were maintained in culture media in 6-well plate for 14 days, fixed with 4% paraformaldehyde, stained with 0.1% crystal violet for colony observation, and counted using light microscope. Each experiment was performed in triplicate and repeated at last three times.

**In vivo metastasis**. The MDA-MB-231-Luc-D3H2LN cell line (MDA-MB-231 cell line engineered to stably express firefly luciferase) (Xenogen Corporation) was infected with lentiviruses carrying ZNF516 or/and EGFR, or lentiviruses carrying control shRNA, ZNF516 shRNA, or/and EGFR shRNA. These cells were inoculated into the left abdominal mammary fat pad ($5 \times 10^6$ cells) of 6-week-old immuno-compromised female SCID beige mice (Charles River, Beijing, China). Mice, according to body equal completely randomized design, were divided into groups each with six before injected. Sample size estimate was based on xenograft assays from literatures. Mice in which tumors did not form would be removed from the study. Tumors were observed in all the mice injected with cells. For biolumine-scence imaging, mice were anesthetized and given 200 μg/g of D-luciferin in PBS by intraperitoneal injection. Fifteen minutes after injection, bioluminescence images were obtained with a charge-coupled device camera (IVIS; Xenogen). Biolumi-nescence from relative optical intensity was defined manually, and data was expressed as photon flux (photons/s/cm²/Steradian) and normalized to background photon flux which was defined from a relative optical intensity drawn over a mouse that was not given an injection of D-luciferin. The measurement and data pro-cessing were done by an investigator blinded to the treatment procedure. All animals were killed at the end of the experiment. Animal handling and procedures were approved by the Peking University Health Science Center Institutional Ani-mal Care and Use Committee (LA2015019).

**Patients and specimens**. The 20 samples of triple-negative breast cancer paired with adjacent normal mammary tissues were obtained from surgical specimens from patients with breast cancer. Samples were selected from patients for whom complete information on clinicopathological characteristics was available. Samples were frozen in liquid nitrogen immediately after surgical removal and maintained at −80 °C until protein extraction. All studies were approved by the Ethics Com-mittee of the Peking University Health Science Center, and informed consent was obtained from all patients.

**Statistical analysis**. Data from biological triplicate experiments are presented with error bar as mean ± S.D. unless otherwise noted. Two-tailed unpaired Student's *t*-test was used for comparing two groups of data unless otherwise noted. Statistical significance was considered at a value of $P < 0.05$. SPSS version 13.0 was used for statistical analysis. Before statistical analysis, variation within each group of data and the assumptions of the tests were checked. The correlation coefficients were calculated by R programming.

**Data availability**. ChIP-seq data for ZNF516 has been deposited in the Gene Expression Omnibus (GEO) with an accession number GSE97962 (https://www.ncbi.nlm.nih.gov/geo/query/acc.cgi?acc=GSE97962). Breast data referenced in this study are available in GEO database (http://www.ncbi.nlm.nih.gov/geo) with the accession code numbers GSE36546, GSE1456, GSE42568, GSE36774, and GSE21653. Data for Kaplan–Meier survival analysis in breast cancer patients are publicly available online at http://kmplot.com/analysis/index.php?p=service&cancer=breast. Data for Gene expression-based Outcome for Breast cancer Online (GOBO) analysis are available from http://co.bmc.lu.se/gobo/gsa.pl. All other remaining data are available within the article and its Supplementary Files, or available from the authors upon request.

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

## Acknowledgements

This work was supported by grants (81372223, 81422034, and 3157080222 to L.S., and 81530073 to Y.S.) from National Natural Science Foundation of China, and a grant (2016YFC1302304 to Y.S.) from the Ministry of Science and Technology of China.

## Author contributions

L.L., J.L., Y.S., and L.S. conceived the project and designed the experiments; L.L., Xinhua, L., L.H., and J.Y. performed experiments and analyzed data; F.P. performed pathologic analysis; L.L., J.Y., and Z.Y. performed animal experiments; W.L., S.L., Z.C., G.X., B.X., X.T., Z.Z., T.J., Xujun, L., W.Z., S.Y., C.W., Y.Z., Xiaohan, Y., and Xia, Y. provided technical assistance; L.L., Y.S., and L.S. wrote the manuscript.

## Additional information

**Competing interests:** The authors declare no competing financial interests.

