## [Peer Review File · Nature Communications]

Reviewers' comments:

Reviewer #1 (Remarks to the Author):

In this manuscript, the authors report that the Krüppel C2H2-type zinc-finger protein ZNF516 is physically associated with the CtBP/LSD1/CoREST transcriptional repressor complex. Genome-wide analysis of the transcriptional targets of the ZNF516-CtBP/LSD1/CoREST complex identified a cohort of genes including EGFR that are critically involved in the proliferation and motility of breast cancer cells. ZNF516 was also shown to target a CCTCC motif in the 5' region of EGFR gene and transcriptionally represses EGFR oncogene. They further demonstrated that the ZNF516 inhibits the proliferation and invasive potential of breast cancer cells in vitro and suppresses breast cancer growth and metastasis in vivo. Finally, the expression level of ZNF516 is negatively correlated with that of EGFR, and low expression of ZNF516 is positively associated with advanced pathological staging and poor patient survival of breast carcinomas. These data indicate that ZNF516 is a transcription repressor of EGFR in breast cancer. Given the importance of EGFR in malignant transformation, the results of this study will definitely enhance the understanding of EGFR-related breast carcinogenesis.

The discovery of ZNF516 as a tumor suppressor in breast cancer is novel. ZNF516 has been reported to be involved in Dupuytren's contracture, congenital vertical talus and other developmental diseases. The role of ZNF516 in cancer progression has not been well documented, although few studies reported loss of copy number and hypermethylation of ZNF516 in some kinds of cancer. Moreover, the finding of interaction of ZNF516 and the CtBP/LSD1/CoREST complex is also a novel discovery, providing a new mechanism of EGFR activation in breast cancer. The authors presented a comprehensive in vitro and in vivo experimental design in this paper. The mechanistic studies in this paper (especially the interaction of ZNF516 and CtBP/LSD1/CoREST complex) are convincing and all the figures are clearly presented. The major deficiencies of this study are related to the biological function of this novel interaction and the significance or impact of this study.

1) The whole study did not specifically address the breast cancer subtypes. Given the fact that EGFR expression is a marker for TNBC or Basal-like breast cancer, focusing on TNBC or basal-like breast cancer seems to be more reasonable.

2) Follow up the first question, the research design in the paper sometime is very confusing. For example, the expression of ZNF516 in MCF7 is very high (Figure6A). However, in some panels, the expression of ZNF516 in MCF7 seems very low (Figure 4A and B). This raised another question of using MCF7 as a model for ectopic expression of ZNF516. The same problem exists for MDA-MB231 cells (Figure 6 and Figure 4).

3) The EGFR western blotting results in Figure6A are questionable. It is not consistent with other published papers. EGFR should be highly expressed in SUM1315 cells. Although MCF10A cell is frequently used as a control cell model in breast cancer study, it is a basal-like cells and EGFR expression in this cell model should be much higher than T47D and MCF7 cells (both are luminal cell models).

4) It is good to show the inverse correlation of ZNF516 and EGFR in some data sets. Have the author examined the TCGA data base, a most commonly used data base? Have you performed the similar studies in different subtype of breast cancer?

5) According to the ChIP-seq and RT-PCR results, ZNF516 suppresses expression of several oncogenes including EGFR, TGF β and ERBB3. This result indicates that EGFR is not the only downstream mediator of ZNF516's tumor suppressor function. However, this notion has not been fully addressed in this paper. Moreover, the authors tried to confirm the EGFR role in ZNF516 function by re-expressing EGFR in cells with ectopic ZNF516 expression. This is not an appropriate

approach since you just cannot control the level of EGFR expression back to the original level.

6) It is great to identify the interaction of ZNF516 and the CtBP/LSD1/CoREST transcriptional repressor complex. However, in the functional study, knockdown of CtBP, LSD1 or CoREST has not been tested for tumor growth and metastasis. For this reviewer, this is much more important than re-expression of EGFR.

7) One of the deficiencies in this study is that the critical domains in ZNF516, CoREST, CtBP1 and 2 that are responsible for the interaction have not been identified.

8) The author stated that finding of ZNF516 represses EGFR through interaction with the CtBP/LSD1/CoREST transcriptional repressor complex highlights the possibility of ZNF516 as a target for breast cancer. Could you please explain how a tumor suppressor serves as a therapeutic target?

9) The information of the tumor samples used in Figure 6b was not provided. Are there all basal like tumor, since they all have high EGFR expression?

Based on above comments, I believe that this manuscript requires a major revision before it can be published in Nature Communication.

Reviewer #2 (Remarks to the Author):

NCOMMS-16-24127

In the manuscript titled "ZNF516 Is a Potent Suppressor of the EGFR Oncogene through Chromatin Targeting of the CtBP/LSD1/CoREST Complex", Sun and colleagues have investigated the role of ZNF516 as suppressor of oncogene EGFR expression. The authors have utilized variety of biochemical and sequencing techniques to show that ZNF516 is in the same complex with CtBP/LSD1/CoREST. Furthermore, it is shown that this complex is involved in the suppression of breast cancer proliferation and invasiveness through the inhibition of EGFR expression. This indicates that the complex might be potential therapeutic target in breast cancer.

Although the manuscript contains interesting observations, the results and conclusions are not novel. Previous studies have already characterized the action of CtBP-complex in detail in breast cancer cells. Furthermore, the previously published results are in direct conflict with the data presented in this manuscript, and the authors don't acknowledge or discuss this discrepancy. In addition, the ChIP sequencing data is insufficiently analyzed with little or no details of analysis parameters. Also the manuscript contains several other issues with analysis and presentation of the data.

The authors would need to indicate in detail the novelty of their finding compared to the published results. Also the discrepancy would need to be addressed in extreme detail. Further, especially the ChIP-sequencing data would need to be reanalyzed and the authors would need to provide more details on the data analysis. My detailed comments:

1.

Novelty of ZNF516 action. The action of ZNF516 seem to be dependent on the action of other components of the repression complex such as CtBP1. Which suggest that the ZNF516 is not a potent suppressor of the EGFR oncogene. Furthermore, there are discrepancies between data shown and published data. The authors indicate that the ZNF516 and CtBP1 bind to the same promoter sites and the anti-proliferative effect of ZNF516 overexpression is blunted by the depletion of CtBP1, LSD1, CoREST. This suggest that ZNF516 might not be needed for EGFR

downregulation. Is ZNF516 needed for repressing EGFR expression in cells with overexpression of CtBP1, LSD1 or CoREST? Further, the authors show that overexpression of ZNF516 leads to increase in E-cadherin and decrease in Vimentin expression. Once ZNF516 is depleted, the expression of Vimentin is increased and E-cadherin decrease. This indicates that epithelial-to-mesenchymal transition (EMT) is regulated by ZNF516 and ZNF516 favors epithelial over mesenchymal markers. It is also shown that depletion of CtBP leads in increase of Vimentin and decrease in E-cadherin expression. This is similar what is seen with ZNF516. However, published data (ref 42) shows the complete opposite effect. In ref 42, depletion of CtBP increases E-cadherin expression while decreasing Vimentin expression. Further, ref 42 show that CtBP regulates mesenchymal markers over epithelial. If ZNF516 is in the same complex with CtBP and CtBP is needed for downregulation of EGFR, how does these factors have opposite effects on EMT in the same cell line? These effect are not discussed by the authors. This is extremely confusing and should be clarified and discussed in extreme detail by the authors. In addition, the authors indicate that the high levels of ZNF516 is associated with higher survival. However, high levels of CtBP (ref 42) is associated with poor survival. Again, if ZNF516 is in the same complex with CtBP and CtBP is needed for downregulation of EGFR, how does these factors have opposite effects for survival (NB! The reviewer is not part or associated with the Kevin Gardner lab).

2.

ChIP-sequencing data. Sufficient information of ChIP-seq samples and data analysis is not provided by the authors. First and foremost, p-value is not an appropriate cutoff in MACS if no false discovery rate (FDR) range is indicated. In MACS each peak in control and treatment conditions is given a p-value. This means that peaks with low p-values in treatment conditions (ZNF516 ChIP-seq) can have high enrichment and low p-value also in control condition (IgG ChIP-seq). By comparing p-value of each individual peak between control and treatment, empirical FDR is assigned to each peak in treatment conditions. This means that FDR indicates the significance of peaks. Either FDR cutoff should be used or FDR-range should be indicated for the peaks with the used p-value cutoff. By using only p-value cutoff, significant number of peaks can be background noise. In addition, several key elements of proper ChIP-seq reporting is not provided in the text, legends or methods by the authors. Specifically:

- Was replicate samples done?
- No quality scores are provided e.g. number of mapped reads.
- No details provided of the downstream analyses e.g. pathway and motif analyses.
- No examples of ChIP-seq tracks are shown. This should at least be shown at EGFR promoter.

Due to these aspects it is extremely difficult to assess the quality and reliability of the ChIP-seq data. Further, the representation and reporting the ChIP-seq data is confusing. Specifically:

- How many binding sites were found in CtBP1 by the authors? Does this number differ from ref 42?
- In ref 42, the CtBP is enriched at ETS, CREB, STAT and SP transcription family motifs at promoters. The motif found by the authors closely resembles the SP1 motif. Are these factors directly binding to DNA or tethering to these factors? How many of the 507 sites harbor this motif? What other motifs were enriched at these sites? No information was provided by the authors in the text or in the methods.
- The authors are only looking cobinding of ZNF516 and CtBP at promoters? Why? Most of the binding of ZNF516 and CtBP1 is at intronic and intergenic regions. This should be clarified.
- Figure 3B does not represent peak numbers as indicated in the text. Number of peaks in each cluster should be indicated.
- The authors indicate that knockdown of ZNF516 and other corepressor components validate the ChIP-seq data. How do these results validate ChIP-seq data? No knockdown or overexpression ChIP-seq experiments were performed. Please clarify.

3.

Other issues.

- Some of the western blots don't have any loading controls and majority of western blots don't have any molecular weight markers. This should be rectified.
- In Figure 2B, how can input be less than the IP? If all ZNF516 is interacting with protein X then input should equal IP. Please clarify.
- Why was the mass spectrometric analysis of ZNF516 interactome done in HEK293T cells and the downstream analyses in breast cancer cells? The authors given no reason for using HEK293T cells in this experiment.
- The experiments in Figure 5 was performed in MDA-MB-231 cells. Why these experiments are not done in the MCF-7 cells which are used in the earlier experiments? The authors given no reason for using MDA-MB-231 cells in this experiment. Please clarify.

Reviewer #3 (Remarks to the Author):

ZNF516 is a Kruppel family zinc finger protein that has been linked to diseases including cancer, where evidence suggests both negative and positive connections. This study investigates biological activities of ZNF516 in transcriptional regulation and in breast cancer cells. In Gal4 transcriptional reporter assays, ZNF516 acts as a repressor. Pull-down/mass spectrometry identified ZNF516 in complex with members of the CtBP/SDK1/CoREST complex, which was verified with co-immunoprecipitations, co-elution in gel filtration. A small deletion/addback pulldown analysis identified interactions of ZNF516 amino terminal domain with CoREST, and the middle domain with CtBP1, and CtBP2. Transcriptional targets were identified for the complex, and inhibitor experiments implicated both histone deacetylase and histone lysine demethylase activities for transcriptional repression in Gal4 reporter assays. ChIP-seq with anti-ZNF516 identified binding sites in MCF-7 cells with significant overlap of a subset with CtBP1 ChIP sites. Both proteins ChIP to targets with CCTCC consensus. A panel of ten target genes including EGFR was validated for ZNF516-dependency of mRNA expression. Knockdown of ZNF516 eliminated CtBP1 ChIP to the targeted promoters and for CoREST ChIP to these sites. Overall, CtBP1 and CoREST are targeted to promoters by ZNF516 for transcriptional repression, histone demethylation, and histone acetylation. In two mammary cell lines, expression of EGFR transcript and protein was inversely affected by ZNF516 knockdown or overexpression, which is associated with ChIP of all four complex components. Complex activity inversely affected cell proliferation, and clonogenic growth, both of which required, in part, EGFR. ZNF516 was inversely associated with EMT, and in an orthotopic mammary fat pad xenograft model, ZNF516 reduced primary and metastatic tumor growth in a partly EGFR-dependent fashion. In human tumor dataset, ZNF516 expression is negatively correlated with EGFR; ZNF516 is inversely associated with higher stage and grade and poorer prognosis.

This study begins with an exceptionally thorough and technically excellent analysis of ZNF516 as a transcriptional regulator, including comprehensive identification of transcriptional targets and ChIP sites in one cell lines, identification of the three other members of this complex as obligatory co-factors for regulation of one panel of target genes, and comprehensive validation of dependencies of complex components with ChIP sites, histone modifications, and regulation of targets. These results are mainly novel, although the association with CtBP1 is reported online. They are consistent with expectations from some other members of this gene family. The regulation of EGFR transcription, and in vitro and xenograft correlates are well-supported. I have some concerns about the impact of EGFR regulation, and the biomarker studies, but the experimental analysis is thoroughly accomplished.

1. In Fig. 4A and 4B, the regulation of EGFR protein levels seems to vary by a factor of only around two with substantial changes in ZNF516 expression. While the result is convincing, it is hard to understand how such small changes in steady-state receptor expression, which presumably represent the range of protein regulation through ZNF516, have such a major impact on

phenotype. With a larger number of other co-regulated genes, and even with the EGFR knockdown and overexpression controls, it is possible that co-regulated proteins are involved. This could be clarified with CRISPR knockouts.

2.EMT. Even in the small panel of target genes chosen, in addition to the EGFR, there are at least two proteins possibly involved in EMT regulation: TGFB3 and SMAD3. Moreover, in mammary cells, induced EMT has been shown to reduce EGFR expression, so the dependent relationships are not clear here. CtBP/LSD1/CoREST complex also associates with EMT promoters ZEB1 and ZEB2, and in the long run it will be interesting to know how these factors interact with ZNF516 in transcriptional regulation.

3.Subtype. Breast cancer prognosis is strongly associated with transcriptional subtype, and it is possible that the biomarker associations are tracking subtype rather than phenotypic effects of ZNF516. For example, EGFR expression is enhanced in basal-like breast cancer (possibly, according to this manuscript through low ZNF516). To understand the prognostic associations of ZNF516, we really need to know its relationship with transcriptional subtype.

minor issue. The working regarding reference 14, line 86 and Discussion, implies that only the regulatory sequence in intron 1 is amplified. This is surprising, if literally interpreted. Was this worded as meant? There should be substantial high resolution data on this amplification by now.

Response to reviewers' comments-

Reviewer #1:

In this manuscript, the authors report that the Krüppel C2H2-type zinc-finger protein ZNF516 is physically associated with the CtBP/LSD1/CoREST transcriptional repressor complex. Genome-wide analysis of the transcriptional targets of the ZNF516-CtBP/LSD1/CoREST complex identified a cohort of genes including EGFR that are critically involved in the proliferation and motility of breast cancer cells. ZNF516 was also shown to target a CCTCC motif in the 5' region of EGFR gene and transcriptionally represses EGFR oncogene. They further demonstrated that the ZNF516 inhibits the proliferation and invasive potential of breast cancer cells in vitro and suppresses breast cancer growth and metastasis in vivo. Finally, the expression level of ZNF516 is negatively correlated with that of EGFR, and low expression of ZNF516 is positively associated with advanced pathological staging and poor patient survival of breast carcinomas. These data indicate that ZNF516 is a transcription repressor of EGFR in breast cancer. Given the importance of EGFR in malignant transformation, the results of this study will definitely enhance the understanding of EGFR-related breast carcinogenesis.

The discovery of ZNF516 as a tumor suppressor in breast cancer is novel. ZNF516 has been reported to be involved in Dupuytren's contracture, congenital vertical talus and other developmental diseases. The role of ZNF516 in cancer progression has not been well documented, although few studies reported loss of copy number and hypermethylation of ZNF516 in some kinds of cancer. Moreover, the finding of interaction of ZNF516 and the CtBP/LSD1/CoREST complex is also a novel discovery, providing a new mechanism of EGFR activation in breast cancer. The authors presented a comprehensive in vitro and in vivo experimental design in this paper. The mechanistic studies in this paper (especially the interaction of ZNF516 and CtBP/LSD1/CoREST complex) are convincing and all the figures are clearly presented. The major deficiencies of this study are related to the biological function of this novel interaction and the significance or impact of this study.

1) The whole study did not specifically address the breast cancer subtypes. Given the fact that EGFR expression is a marker for TNBC or basal-like breast cancer, focusing on TNBC or basal-like breast cancer seems to be more reasonable.

Authors: Transcriptional repression of EGFR is observed in our study in both MDA-MB-231 (TNBC) and MCF-7 (luminal) cell lines. Although our current focus is the mechanistic role of ZNF516 in breast cancer, future studies need focuses specifically on TNBC or basal-like breast cancer due to their high expression of EGFR, as the reviewer rightfully pointed out.

2) Follow up the first question, the research design in the paper sometime is very confusing. For example, the expression of ZNF516 in MCF7 is very high (Figure 6A). However, in some panels, the expression of ZNF516 in MCF7 seems very low (Figure 4A and B). This raised another question of using MCF7 as a model for ectopic expression of ZNF516. The same problem exists for MDA-MB231 cells (Figure 6 and Figure 4).

Authors: As the Figure numbers indicate, these blots were from different experiments. Due to differences in factors such as protein loading and exposure time in different experiments, it is inappropriate to make comparisons between different blots, especially blots from different experiments.

ZNF516 is expressed in both MCF-7 and MDA-MB-231 cells, though relative higher protein level in MCF-7 than MDA-MB231 cells. Therefore, overexpression and knockdown of ZNF516 were performed in both cell lines in Fig. 4a. To keep pace with Fig. 4b-d, the corresponding experiments have been performed in MDA-MB-231 cells, and the data has been added to revision in Supplementary Fig. S2a-c.

3) The EGFR western blotting results in Figure 6A are questionable. It is not consistent with other published papers. EGFR should be highly expressed in SUM1315 cells. Although MCF10A cell is frequently used as a control cell model in breast cancer study, it is a basal-like cells and EGFR expression in this cell model should be much higher than T47D and MCF7 cells (both are luminal cell models).

Authors: We appreciate the reviewer for pointing this out. We have discussed this issue and repeated the experiment, and the blots have been replaced.

4) It is good to show the inverse correlation of ZNF516 and EGFR in some data sets. Have the author examined the TCGA data base, a most commonly used data base? Have you performed the similar studies in different subtype of breast cancer?

Authors: The correlation of ZNF516 and EGFR has been analyzed in the TCGA database as suggested. The data have been added to revision in Figure 6d. As TCGA database do not have information on molecular subtypes, the analysis on correlation between ZNF516 and EGFR was performed with published clinical datasets (GSE21653), and negative correlations of mRNA levels between ZNF516 and EGFR were found in not only basal-like but also luminal A, luminal B, and HER2-enriched breast cancers. The data have been added to revision as Supplementary Fig. S4c.

5) According to the ChIP-seq and RT-PCR results, ZNF516 suppresses expression of several oncogenes including EGFR, TGFBRb and ERBB3. This result indicates that EGFR is not the only downstream mediator of ZNF516's tumor suppressor function. However, this notion has

not been fully addressed in this paper. Moreover, the authors tried to confirm the EGFR role in ZNF516 function by re-expressing EGFR in cells with ectopic ZNF516 expression. This is not an appropriate approach since you just cannot control the level of EGFR expression back to the original level.

Authors: Our main argument is that ZNF516 is physically associated with the CtBP/LSD1/CoREST transcriptional repressor complex and transcriptionally represses downstream target genes, including, but not limited to, EGFR. We are by no means to exclude the involvement and the importance of other downstream target genes. However, due to technical difficulties, it is impossible to investigate multiple targets in a single project, as the reviewer would probably agree. This point is further emphasized in the revision.

For the re-expressing EGFR experiments, the level of EGFR expression was controlled by creating stable clones of cells that were expressing different levels of EGFR, and the clones with EGFR levels close to original EGFR level were chosen for phenotype experiments. The details have been added to the revision in Methods and the data have been added to the revision in Supplementary Fig. S3a.

6) It is great to identify the interaction of ZNF516 and the CtBP/LSD1/CoREST transcriptional repressor complex. However, in the functional study, knockdown of CtBP, LSD1 or CoREST has not been tested for tumor growth and metastasis. For this reviewer, this is much more important than re-expression of EGFR.

Authors: To comply with the reviewer's requests, we have performed the suggested experiments and added the data to the revision as Fig. 5c,g and Supplementary Fig. S3b-d.

7) One of the deficiencies in this study is that the critical domains in ZNF516, CoREST, CtBP1 and 2 that are responsible for the interaction have not been identified.

Authors: The suggested experiments have been performed and data have been added to the revision as Fig. 2g.

8) The author stated that finding of ZNF516 represses EGFR through interaction with the CtBP/LSD1/CoREST transcriptional repressor complex highlights the possibility of ZNF516 as a target for breast cancer. Could you please explain how a tumor suppressor serves as a therapeutic target?

Authors: Tumor suppressors have been pursued as therapeutic targets, exemplified by Gendicin, the first adenoviral p53-based gene therapy¹ and by recombinant adenovirus-PTEN to restore tumor suppressor PTEN in multiple myeloma tumor cells². Obviously, the significance of ZNF516 is not comparable to that of p53 and PTEN, but the possibility exists.

9) The information of the tumor samples used in Figure 6b was not provided. Are there all basal like tumor, since they all have high EGFR expression?

Authors: The subtype of tumor samples used in Fig. 6b is all triple-negative breast cancer. The description has been added to the revision.

References to Reviewer #1:

1. Peng Z. Current status of gendicine in China: recombinant human Ad-p53 agent for treatment of cancers. *Hum Gene Ther* **16**, 1016-1027 (2005).
2. Wang S, *et al.* Effect of wild type PTEN gene on proliferation and invasion of multiple myeloma. *Int J Hematol* **92**, 83-94 (2010).

RE: NCOMMS-16-24127

Response to reviewers' comments-

Reviewer #2:

In the manuscript titled “ZNF516 Is a Potent Suppressor of the EGFR Oncogene through Chromatin Targeting of the CtBP/LSD1/CoREST Complex”, Sun and colleagues have investigated the role of ZNF516 as suppressor of oncogene EGFR expression. The authors have utilized a variety of biochemical and sequencing techniques to show that ZNF516 is in the same complex with CtBP/LSD1/CoREST. Furthermore, it is shown that this complex is involved in the suppression of breast cancer proliferation and invasiveness through the inhibition of EGFR expression. This indicates that the complex might be a potential therapeutic target in breast cancer.

Although the manuscript contains interesting observations, the results and conclusions are not novel. Previous studies have already characterized the action of CtBP-complex in detail in breast cancer cells. Furthermore, the previously published results are in direct conflict with the data presented in this manuscript, and the authors don't acknowledge or discuss this discrepancy. In addition, the ChIP sequencing data is insufficiently analyzed with little or no details of analysis parameters. Also the manuscript contains several other issues with analysis and presentation of the data.

The authors would need to indicate in detail the novelty of their finding compared to the published results. Also the discrepancy would need to be addressed in extreme detail. Further, especially the ChIP-sequencing data would need to be reanalyzed and the authors would need to provide more details on the data analysis. My detailed comments:

1. Novelty of ZNF516 action. The action of ZNF516 seems to be dependent on the action of other components of the repression complex such as CtBP1. Which suggests that the ZNF516 is not a potent suppressor of the EGFR oncogene. Furthermore, there are discrepancies between data shown and published data. The authors indicate that the ZNF516 and CtBP1 bind to the same promoter sites and the anti-proliferative effect of ZNF516 overexpression is blunted by the depletion of CtBP1, LSD1, CoREST. This suggests that ZNF516 might not be needed for EGFR downregulation. Is ZNF516 needed for repressing EGFR expression in cells with overexpression of CtBP1, LSD1 or CoREST?

Authors: We appreciate the concerns raised by the reviewer. CtBP1, LSD1, and CoREST all represent corepressors or corepressor complexes, meaning that these proteins or protein complexes do not have the ability to bind to DNA^{1,2,3}. Our observations that ZNF516, by its name a zinc finger-containing protein that is capable of DNA binding, is physically associated

with CtBP1, LSD1, and CoREST and acts as a repressor (not corepressor) provide a molecular basis for the recruitment and functions of CtBP1, LSD1, and CoREST to/on chromatin. These findings are important and are novel. Moreover, as agreed to by the other two reviewers, our finding that ZNF516 acts as a potential tumor suppressor in breast cancer is also significant and novel.

In responding to the reviewer requests, CtBP1, LSD1 or CoREST was overexpressed in MCF-7 cells, and the expression of EGFR was examined. The results showed that overexpression of CtBP1, LSD1 or CoREST resulted in a reduced expression of EGFR protein. However, when ZNF516 was depleted, overexpression of CtBP1, LSD1 or CoREST no longer repressed the expression of EGFR. These results are consistent with the argument provided above and in our manuscript. The data have been provided in Data to the Reviewers as Fig. R2. In addition, our experiments in Fig. 3h also support a notion that ZNF516 is required for the recruitment of CtBP1 and CoREST on target gene promoters in transcription repression of target genes.

Further, the authors show that overexpression of ZNF516 leads to increase in E-cadherin and decrease in Vimentin expression. Once ZNF516 is depleted, the expression of Vimentin is increased and E-cadherin decrease. This indicates that epithelial-to-mesenchymal transition (EMT) is regulated by ZNF516 and ZNF516 favors epithelial over mesenchymal markers. It is also shown that depletion of CtBP leads in increase of Vimentin and decrease in E-cadherin expression. This is similar what is seen with ZNF516. However, published data (ref 42) shows the complete opposite effect. In ref 42, depletion of CtBP increases E-cadherin expression while decreasing Vimentin expression. Further, ref 42 show that CtBP regulates mesenchymal markers over epithelial. If ZNF516 is in the same complex with CtBP and CtBP is needed for downregulation of EGFR, how does these factors have opposite effects on EMT in the same cell line? These effect are not discussed by the authors. This is extremely confusing and should be clarified and discussed in extreme detail by the authors.

Authors: As mentioned above, CtBP is a corepressor protein lacking of the ability to bind DNA. Its actions and cellular readouts depend on its recruitment to chromatin by various transcription factors capable of DNA binding^{3,4}. Thus, recruitment of CtBP, via its PLDLS binding cleft, by different transcription factors could elicit different cellular outputs. We appreciate the reviewer for this point and discussed the discrepancies.

In addition, the authors indicate that the high levels of ZNF516 is associated with higher survival. However, high levels of CtBP (ref 42) is associated with poor survival. Again, if ZNF516 is in the same complex with CtBP and CtBP is needed for downregulation of EGFR, how does these factors have opposite effects for survival (NB! The reviewer is not part or associated with the Kevin Gardner lab).

Authors: As stated above, CtBP could be recruited to chromatin by different repressors and generates different cellular outputs. At least in our study, Kaplan-Meier survival analysis showed that higher CtBP1 expression is associated with a better relapse-free survival of breast cancer patients ($n = 3951$) ($p = 7.5e-08$). The data have been provided in the revision in Fig. 6e. Using TCGA database, we found a positive correlation in the mRNA expression of ZNF516 and CtBP1, and the data have been provided in the revision in Fig. 6d. Further stratification of patient groups based on the coexpression of ZNF516 and CtBP1 improved the predictive capability of ZNF516 ($p = 0.0064$) (GSE1456). These data have been provided in the revision in Fig. 6f.

2. ChIP-sequencing data. Sufficient information of ChIP-seq samples and data analysis is not provided by the authors. First and foremost, p-value is not an appropriate cutoff in MACS if no false discovery rate (FDR) range is indicated. In MACS each peak in control and treatment conditions is given a p-value. This means that peaks with low p-values in treatment conditions (ZNF516 ChIP-seq) can have high enrichment and low p-value also in control condition (IgG ChIP-seq). By comparing p-value of each individual peak between control and treatment, empirical FDR is assigned to each peak in treatment conditions. This means that FDR indicates the significance of peaks. Either FDR cutoff should be used or FDR-range should be indicated for the peaks with the used p-value cutoff. By using only p-value cutoff, significant number of peaks can be background noise. In addition, several key elements of proper ChIP-seq reporting is not provided in the text, legends or methods by the authors.

Authors: According to README for MACS (Version2.1.0) (<https://github.com/taoliu/MACS>)⁵, if p -value is specified, MACS should use p -value instead of q -value. Moreover, a relatively stringent threshold, p -value $< 10^{-3}$, was used in the current study as well as in other studies in our lab^{6,7}, which is helpful in filtering background noise.

Specifically:

- Was replicate samples done?

Authors: ChIP-seq samples in our lab are typically subjected to strict quality control by the sequencing company (BGI). Due to the expense and the time, the samples are not replicated.

- No quality scores are provided e.g. number of mapped reads.

Authors: The number of mapped reads of ZNF516 ChIP and input has been provided in the revision as suggested.

- No details provided of the downstream analyses e.g. pathway and motif analyses.

Authors: Pathways with their gene hits and p -value (represented as $-\log_{10}(p\text{-value})$) have been provided in the Supplementary Table S2. The top 10 motifs enriched at ZNF516 peaks via MEME-ChIP analysis and their corresponding E-values have been provided in the Supplementary Fig. S1.

- No examples of ChIP-seq tracks are shown. This should at least be shown at EGFR promoter.

Authors: The ChIP-seq track at EGFR promoter has been added to the revision as Fig. 4f.

Due to these aspects it is extremely difficult to assess the quality and reliability of the ChIP-seq data. Further, the representation and reporting the ChIP-seq data is confusing. Specifically:

- How many binding sites were found in CtBP1 by the authors? Does this number differ from ref 42?

Authors: A total of 38,944 CtBP1 specific binding sites were obtained in this study and this number differs from that in ref 42, possibly due to difference in peak calling.

- In ref 42, the CtBP is enriched at ETS, CREB, STAT and SP transcription family motifs at promoters. The motif found by the authors closely resembles the SP1 motif. Are these factors directly binding to DNA or tethering to these factors?

Authors: As transcription factors, ETS, CREB, STAT and SP are all capable of directly binding to DNA and recruiting cofactors such as CtBP. The binding motif of ZNF516 CCTCC indeed closely resembles the SP1 recognition sequence, suggesting that SP1 and ZNF516 might competitively interact with CtBP complex to regulate certain target genes. Consistently, SP1 has been reported to activate EGFR transcription⁸ and promote the proliferation of breast cancer cells^{9,10}, supporting our argument that ZNF516 is a transcription repressor and has the opposite effects on the transcriptional regulation with SP1.

- How many of the 507 sites harbor this motif? What other motifs were enriched at these sites? No information was provided by the authors in the text or in the methods.

Authors: We searched the presence of CCTCC within the 507 genes through Find Motif Occurrence (FIMO) scanner¹¹ and found that 53% of the peaks harbor this motif. The top 10 motifs enriched at ZNF516 peaks via MEME-ChIP analysis were provided in the Supplementary Fig. S1 and related information has been added to the text.

- The authors are only looking cobinding of ZNF516 and CtBP at promoters? Why? Most of the binding of ZNF516 and CtBP1 is at intronic and intergenic regions. This should be clarified.

Authors: Since transcription factors typically recognize and bind to specific DNA sequences in the promoter of target genes via characteristic DNA-binding domains^{12, 13}, peaks on promoters are preferentially analyzed in the study for transcription regulation. This has been clarified in the revision.

- Figure 3B does not represent peak numbers as indicated in the text. Number of peaks in each cluster should be indicated.

Authors: Number of peaks in each cluster has been indicated in Fig. 3b according to the suggestions.

- The authors indicate that knockdown of ZNF516 and other corepressor components validate the ChIP-seq data. How do these results validate ChIP-seq data? No knockdown or overexpression ChIP-seq experiments were performed. Please clarify.

Authors: We have modified the text and apologize for the confusion.

3. Other issues.

1- Some of the western blots don't have any loading controls and majority of western blots don't have any molecular weight markers. This should be rectified.

Authors: The loading controls and molecular weight markers have been added to the revision as suggested.

2- In Figure 2B, how can input be less than the IP? If all ZNF516 is interacting with protein X then input should equal IP. Please clarify.

Authors: For immunoprecipitation assays, 5-20% cellular extracts were generally used for input^{14, 15, 16, 17}. The input amount used in our immunoprecipitation assays is 5% (1:20). The information has been added to Methods in the revision.

3- Why was the mass spectrometric analysis of ZNF516 interactome done in HEK293T cells and the downstream analyses in breast cancer cells? The authors given no reason for using HEK293T cells in this experiment.

Authors: HEK293T cells embody some principal attributes for mass spectrometric analysis of protein interactome^{18, 19, 20, 21, 22}, in which the large amounts of protein were required: quick and easy reproduction and maintenance, high efficiency of transfection and protein production; and faithful translation and processing of proteins²³.

4- The experiments in Figure 5 was performed in MDA-MB-231 cells. Why these experiments are not done in the MCF-7 cells which are used in the earlier experiments? The authors given no reason for using MDA-MB-231 cells in this experiment. Please clarify.

Authors: The MDA-MB-231 cells are highly metastatic and are extensively used as a well-established model for matrigel invasion assay and in vivo metastasis assays^{24,25}.

References for Reviewer #2:

1. Nibu Y, Zhang H, Levine M. Interaction of short-range repressors with Drosophila CtBP in the embryo. *Science* **280**, 101-104 (1998).
2. Furusawa T, Moribe H, Kondoh H, Higashi Y. Identification of CtBP1 and CtBP2 as corepressors of zinc finger-homeodomain factor deltaEF1. *Mol Cell Biol* **19**, 8581-8590 (1999).
3. Turner J, Crossley M. Cloning and characterization of mCtBP2, a co-repressor that associates with basic Kruppel-like factor and other mammalian transcriptional regulators. *EMBO J* **17**, 5129-5140 (1998).
4. Kuppuswamy M, *et al.* Role of the PLDLS-binding cleft region of CtBP1 in recruitment of core and auxiliary components of the corepressor complex. *Mol Cell Biol* **28**, 269-281 (2008).
5. Feng J, Liu T, Qin B, Zhang Y, Liu XS. Identifying ChIP-seq enrichment using MACS. *Nat Protoc* **7**, 1728-1740 (2012).
6. Si W, *et al.* Dysfunction of the Reciprocal Feedback Loop between GATA3- and ZEB2-Nucleated Repression Programs Contributes to Breast Cancer Metastasis. *Cancer cell* **27**, 822-836 (2015).
7. Shan L, *et al.* FOXK2 Elicits Massive Transcription Repression and Suppresses the Hypoxic Response and Breast Cancer Carcinogenesis. *Cancer cell* **30**, 708-722 (2016).
8. Kageyama R, Merlino GT, Pastan I. Epidermal growth factor (EGF) receptor gene transcription. Requirement for Sp1 and an EGF receptor-specific factor. *J Biol Chem* **263**, 6329-6336 (1988).
9. Wang Y, *et al.* HBXIP up-regulates ACSL1 through activating transcriptional factor

- Sp1 in breast cancer. *Biochem Biophys Res Commun* **484**, 565-571 (2017).
10. Zhang Y, *et al.* The oncoprotein HBXIP upregulates PDGFB via activating transcription factor Sp1 to promote the proliferation of breast cancer cells. *Biochem Biophys Res Commun* **434**, 305-310 (2013).
 11. Grant CE, Bailey TL, Noble WS. FIMO: scanning for occurrences of a given motif. *Bioinformatics* **27**, 1017-1018 (2011).
 12. Kornberg RD. The molecular basis of eukaryotic transcription. *Proc Natl Acad Sci U S A* **104**, 12955-12961 (2007).
 13. Maston GA, Evans SK, Green MR. Transcriptional regulatory elements in the human genome. *Annu Rev Genomics Hum Genet* **7**, 29-59 (2006).
 14. Lan F, *et al.* Recognition of unmethylated histone H3 lysine 4 links BHC80 to LSD1-mediated gene repression. *Nature* **448**, 718-722 (2007).
 15. Han X, *et al.* Destabilizing LSD1 by Jade-2 promotes neurogenesis: an antibraking system in neural development. *Mol Cell* **55**, 482-494 (2014).
 16. Wang Y, *et al.* LSD1 is a subunit of the NuRD complex and targets the metastasis programs in breast cancer. *Cell* **138**, 660-672 (2009).
 17. Zhang J, *et al.* SFMBT1 functions with LSD1 to regulate expression of canonical histone genes and chromatin-related factors. *Genes Dev* **27**, 749-766 (2013).
 18. Lee MG, Wynder C, Cooch N, Shiekhattar R. An essential role for CoREST in nucleosomal histone 3 lysine 4 demethylation. *Nature* **437**, 432-435 (2005).
 19. Lee MG, Wynder C, Bochar DA, Hakimi MA, Cooch N, Shiekhattar R. Functional interplay between histone demethylase and deacetylase enzymes. *Mol Cell Biol* **26**, 6395-6402 (2006).
 20. Tang M, *et al.* The malignant brain tumor (MBT) domain protein SFMBT1 is an integral histone reader subunit of the LSD1 demethylase complex for chromatin association and epithelial-to-mesenchymal transition. *J Biol Chem* **288**, 27680-27691 (2013).
 21. Xing M, *et al.* Interactome analysis identifies a new paralogue of XRCC4 in non-homologous end joining DNA repair pathway. *Nat Commun* **6**, 6233 (2015).

22. Yi T, *et al.* eIF1A augments Ago2-mediated Dicer-independent miRNA biogenesis and RNA interference. *Nat Commun* **6**, 7194 (2015).
23. Thomas P, Smart TG. HEK293 cell line: a vehicle for the expression of recombinant proteins. *J Pharmacol Toxicol Methods* **51**, 187-200 (2005).
24. Clarke R. Human breast cancer cell line xenografts as models of breast cancer. The immunobiologies of recipient mice and the characteristics of several tumorigenic cell lines. *Breast Cancer Res Treat* **39**, 69-86 (1996).
25. Price JE, Polyzos A, Zhang RD, Daniels LM. Tumorigenicity and metastasis of human breast carcinoma cell lines in nude mice. *Cancer Res* **50**, 717-721 (1990).

Response to reviewers' comments-

Reviewer #3:

ZNF516 is a Kruppel family zinc finger protein that has been linked to diseases including cancer, where evidence suggests both negative and positive connections. This study investigates biological activities of ZNF516 in transcriptional regulation and in breast cancer cells. In Gal4 transcriptional reporter assays, ZNF516 acts as a repressor. Pull-down/mass spectrometry identified ZNF516 in complex with members of the CtBP/SDK1/CoREST complex, which was verified with co-immunoprecipitations, co-elution in gel filtration. A small deletion/addback pulldown analysis identified interactions of ZNF516 amino terminal domain with CoREST, and the middle domain with CtBP1, and CtBP2. Transcriptional targets were identified for the complex, and inhibitor experiments implicated both histone deacetylase and histone lysine demethylase activities for transcriptional repression in Gal4 reporter assays. ChIP-seq with anti-ZNF516 identified binding sites in MCF-7 cells with significant overlap of a subset with CtBP1 ChIP sites. Both proteins ChIP to targets with CCTCC consensus. A panel of ten target genes including EGFR was validated for ZNF516-dependency of mRNA expression. Knockdown of ZNF516 eliminated CtBP1 ChIP to the targeted promoters and for CoREST ChIP to these sites. Overall, CtBP1 and CoREST are targeted to promoters by ZNF516 for transcriptional repression, histone demethylation, and histone acetylation. In two mammary cell lines, expression of EGFR transcript and protein was inversely affected by ZNF516 knockdown or overexpression, which is associated with ChIP of all four complex components. Complex activity inversely affected cell proliferation, and clonogenic growth, both of which required, in part, EGFR. ZNF516 was inversely associated with EMT, and in an orthotopic mammary fat pad xenograft model, ZNF516 reduced primary and metastatic tumor growth in a partly EGFR-dependent fashion. In human tumor datasets, ZNF516 expression is negatively correlated with EGFR; ZNF516 is inversely associated with higher stage and grade and poorer prognosis.

This study begins with an exceptionally thorough and technically excellent analysis of ZNF516 as a transcriptional regulator, including comprehensive identification of transcriptional targets and ChIP sites in one cell lines, identification of the three other members of this complex as obligatory co-factors for regulation of one panel of target genes, and comprehensive validation of dependencies of complex components with ChIP sites, histone modifications, and regulation of targets. These results are mainly novel, although the association with CtBP1 is reported online. They are consistent with expectations from some other members of this gene family. The regulation of EGFR transcription, and in vitro and xenograft correlates are well-supported. I have some concerns about the impact of EGFR regulation, and the biomarker studies, but the experimental analysis is thoroughly accomplished.

1. In Fig. 4A and 4B, the regulation of EGFR protein levels seems to vary by a factor of only around two with substantial changes in ZNF516 expression. While the result is convincing, it is hard to understand how such small changes in steady-state receptor expression, which presumably represent the range of protein regulation through ZNF516, have such a major impact on phenotype. With a larger number of other co-regulated genes, and even with the EGFR knockdown and overexpression controls, it is possible that co-regulated proteins are involved. This could be clarified with CRISPR knockouts.

Authors: Our main argument is that ZNF516 is physically associated with the CtBP/LSD1/CoREST transcriptional repressor complex and transcriptionally represses downstream target genes, including, but not limited to, EGFR. We are by no means to exclude the involvement and the importance of other downstream target genes. However, due to technical difficulties, it is impossible to investigate multiple targets in a single project, as the reviewer would probably agree. This point is further emphasized in the revision.

2. EMT. Even in the small panel of target genes chosen, in addition to the EGFR, there are at least two proteins possibly involved in EMT regulation: TGFB3 and SMAD3. Moreover, in mammary cells, induced EMT has been shown to reduce EGFR expression, so the dependent relationships are not clear here.

Authors: We appreciate the reviewer for this point. We are aware of the involvement of TGFB3 and SMAD3 in the regulation of EMT. Again, our current study focuses on EGFR and did not mean to exclude TGFB3, SMAD3 and probably other targets.

We are not aware the literatures concerning the changes of EGFR expression during EMT process. Our observation that ZNF516, through repressing downstream target genes including EGFR, inhibits EMT is consistent with the literatures about the role of EGFR in the regulation of EMT. It is possible that the reviewer mentioned reduction of EGFR in induced EMT is a result of some kind feedback regulation. Future investigations will look into this issue.

CtBP/LSD1/CoREST complex also associates with EMT promoters ZEB1 and ZEB2, and in the long run it will be interesting to know how these factors interact with ZNF516 in transcriptional regulation.

Authors: We appreciate and agree to the reviewer on this point, especially since our affinity purification detected the presence of ZEB2 in the interactome of ZNF516.

3. Subtype. Breast cancer prognosis is strongly associated with transcriptional subtype, and it is possible that the biomarker associations are tracking subtype rather than phenotypic effects of ZNF516. For example, EGFR expression is enhanced in basal-like breast cancer (possibly,

according to this manuscript through low ZNF516). To understand the prognostic associations of ZNF516, we really need to know its relationship with transcriptional subtype.

Authors: We fully agree to the reviewer on this point. In response to reviewer's comment, Kaplan-Meier survival analysis (<http://kmplot.com/analysis/>) of public datasets was performed, and we found that higher ZNF516 expression is associated with a better relapse-free survival of breast cancer patients in Luminal A and even in basal-like subtypes. The data has been added to the revision in Supplementary Fig. S4d. Moreover, interrogation of public datasets for ZNF516 or EGFR expression using the Gene expression-based Outcome for Breast cancer Online (GOBO) tool (<http://co.bmc.lu.se/gobo/gsa.pl>) revealed that expression of ZNF516 is higher in Luminal A and Luminal B subtypes and lower in HER-enriched and Basal-like subtypes, where the expression of EGFR is high. The data has been added to the revision in Supplementary Fig. S4a. These analyses support the pursuit of ZNF516 as a potential biomarker for basal-like breast carcinoma.

minor issue. The working regarding reference 14, line 86 and Discussion, implies that only the regulatory sequence in intron 1 is amplified. This is surprising, if literally interpreted. Was this worded as meant? There should be substantial high resolution data on this amplification by now.

Authors: The reference demonstrated that the transcriptional level of EGFR in breast cancer could be regulated by the length of the CA amplification in intron 1 of *EGFR* gene. Recently, direct sequencing of the EGFR PCR products of (CA)_n repeat polymorphism in intron 1 was performed in breast cancer patients^{1,2}. The most frequent allele was (CA)₁₆, which occurred in 46.3% of patients, followed by 20 CA repetitions in 30%, and 18 allele repeats in 12.5%. Moreover, women with shorter CA repeat (<19) are at a significantly higher risk of breast cancer, and much greater risk of developing cancer before the age of 55¹. The combined presence of both longer EGFR (CA)_n polymorphisms and Lys allele of R497K resulted in lower tumor mode metastasis in breast cancer².

References for Reviewer #3:

1. Jami MS, Hemati S, Salehi Z, Tavassoli M. Association between the length of a CA dinucleotide repeat in the EGFR and risk of breast cancer. *Cancer Invest* **26**, 434-437 (2008).
2. Leite MS, *et al.* Epidermal growth factor receptor gene polymorphisms are associated with prognostic features of breast cancer. *BMC Cancer* **14**, 190 (2014).

Reviewers' comments:

Reviewer #1 (Remarks to the Author):

In this revised manuscript, the authors addressed all of the questions I raised. The study is novel and technically excellent. The results about the mechanism studies are convincing and the presentation is outstanding. I have no further questions.

Reviewer #2 (Remarks to the Author):

NCOMMS-16-24127A

In the revised manuscript titled "ZNF516 Is a Potent Suppressor of the EGFR Oncogene through Chromatin Targeting of the CtBP/LSD1/CoREST Complex", Sun and colleagues have managed to address some but not all issues raised by this reviewer. See details below. If these issues are not rectified, I cannot recommend the manuscript to be published in Nature Communications.

1.

Again, first and foremost, unadjusted p-value is not an appropriate cutoff in ChIP-seq peak detection. Even though MACS2 has the option of selecting p-value instead of q-values, it does not indicate that p-value should be used. The q-value is an adjusted p-value, taking in to account the false discovery rate (FDR, Type I error). FDR has a clear, easily understandable meaning, unlike p-value which can be meaningless in this context.

An FDR-adjusted p-value (aka a q-value) of 0.05 implies that we are willing to accept that 5% of the tests found to be statistically significant (by p-value) will be false positives. In simple terms, if you have 100 data points and you use p-value cutoff of 0.05, your error rate is 0.994 $[1-(1-0.05)^{100}]$. This indicates that you have 99.4% change of observing at least one significant results. To know how many of the significant results are false positive we need to adjust the p-value. If we use adjusted p-value of 0.05 your false positive error rate will be 0.0487 $[1-(1-(0.05/100))^{100}]$. This indicates that you have only 4.87% change of observing false positive result. In ChIP-seq, input or control IgG sample is used to estimate the FDR (or q-value) as the number of peaks found in randomized data (input or IgG) versus the number of peaks found in the actual sample. Hence, using only p-value as cutoff can results in situation where you find 40 000 peaks but you don't know how many of these peaks can be false positive. You could find 20 000 similar peaks from input or IgG sample which would indicate that 50% of peaks are false positive. Hence, reporting FDR (q-value) is of outmost importance. FDR (or q-value) of 0.05 is the most commonly accepted value for peaks of good quality (Landt et al. 2012 Genome Res, Feng et al. 2012 Nat Protoc, Shin et al. 2013 Quant Biol).

Because of inappropriate use of statistical methods there is clear difference between the peak numbers obtained by the authors from published data. Using ref 42 (ref 44 in revised manuscript) CtBP data the authors find ~39 000 peaks (using p-value cutoff by MACS2) which is 6-fold more peaks (~6 700) than in the ref 42 article (using FDR cutoff by Hotspots). The difference cannot solely be because of different peak calling algorithms. If authors were to utilize q-value instead of p-value this problem would be solved.

Due to the above-mentioned facts, the authors should use FDR (q-value) cutoff. At minimum, the authors should indicate what is the range of FDR values when they use p-value cutoff.

2.

The authors focus on promoter binding because according to published data (ref 45 and 46) most TFs bind to promoter regions. These references are over 10 years old and since then there has been multiple studies indicating for several TFs that enhancers regulating gene expression are kbs

away from the TSS (e.g. ENCODE consortium publications). The authors own data shows that majority of ZNF516 binding is intronic or intergenic not in the promoter region. It is therefore puzzling why the authors are focusing on promoter sites. The authors should indicate why they focused on promoter sites. The argument based on 10-year-old publications is not a strong one.

In addition, there seems to be different in the number of promoter sites shown in 3b versus 3c. In 3b there are 5607 ZNF516 promoter sites versus 2993 in 3c (2486+507). Similar difference can be seen with CtBP1. This issue should be clarified.

3.

The authors indicate that: "ChIP-seq samples in our lab are typically subjected to strict quality control by the sequencing company (BGI). Due to the expense and the time, the samples are not replicated.". Replicate sequencing samples are needed to control the biological phenomena not the technical aspects of sequencing. If authors are not willing to perform replicate samples, it should be indicated clearly in the results and in the methods that the results are obtained from a single sequencing sample.

4.

Method section lack details of ChIP-seq data analysis. The authors indicate that Bowtie, MACS2 and ChIPseeker were used without providing any other details. How was the data aligned with Bowtie? How many mismatches were allowed? What is the number of reporter alignments? What parameters was used to build the model in MACS2? Were default settings used? What parameters were used with ChIPseeker?

Other minor issues.

- Figure 4f; what peak in the EGFR promoter represent binding sites assessed in the study? This should be indicated in the figure
- The reference for MACS2 program in the methods is ref 47 which the reference for FIMO

Reviewer #3 (Remarks to the Author):

The authors have addressed my major concerns, especially with inclusion of new supplementary data with breakdowns by breast CA subtype.

Response to reviewers' comments-

Reviewer #2:

In the revised manuscript titled “ZNF516 Is a Potent Suppressor of the EGFR Oncogene through Chromatin Targeting of the CtBP/LSD1/CoREST Complex”, Sun and colleagues have managed to address some but not all issues raised by this reviewer. See details below. If these issues are not rectified, I cannot recommend the manuscript to be published in Nature Communications.

1. Again, first and foremost, unadjusted p-value is not an appropriate cutoff in ChIP-seq peak detection. Even though MACS2 has the option of selecting p-value instead of q-values, it does not indicate that p-value should be used. The q-value is an adjusted p-value, taking in to account the false discovery rate (FDR, Type I error). FDR has a clear, easily understandable meaning, unlike p-value which can be meaningless in this context.

An FDR-adjusted p-value (aka a q-value) of 0.05 implies that we are willing to accept that 5% of the tests found to be statistically significant (by p-value) will be false positives. In simple terms, if you have 100 data points and you use p-value cutoff of 0.05, your error rate is 0.994 $[1-(1-0.05)^{100}]$. This indicates that you have 99.4% chance of observing at least one significant results. To know how many of the significant results are false positive we need to adjust the p-value. If we use adjusted p-value of 0.05 your false positive error rate will be 0.0487 $[1-(1-(0.05/100))^{100}]$. This indicates that you have only 4.87% chance of observing false positive result. In ChIP-seq, input or control IgG sample is used to estimate the FDR (or q-value) as the number of peaks found in randomized data (input or IgG) versus the number of peaks found in the actual sample. Hence, using only p-value as cutoff can result in situation where you find 40 000 peaks but you don't know how many of these peaks can be false positive. You could find 20 000 similar peaks from input or IgG sample which would indicate that 50% of peaks are false positive. Hence, reporting FDR (q-value) is of utmost importance. FDR (or q-value) of 0.05 is the most commonly accepted value for peaks of good quality (Landt et al. 2012 Genome Res, Feng et al. 2012 Nat Protoc, Shin et al. 2013 Quant Biol).

Because of inappropriate use of statistical methods there is clear difference between the peak numbers obtained by the authors from published data. Using ref 42 (ref 44 in revised manuscript) CtBP data the authors find ~39 000 peaks (using p-value cutoff by MACS2) which is 6-fold more peaks (~6 700) than in the ref 42 article (using FDR cutoff by Hotspots). The difference cannot solely be because of different peak calling algorithms. If authors were to utilize q-value instead of p-value this problem would be solved.

Due to the above-mentioned facts, the authors should use FDR (q-value) cutoff. At minimum, the authors should indicate what is the range of FDR values when they use p-value cutoff.

Authors: To comply with the reviewer's request, we have reanalyzed the raw data with a FDR cutoff of 0.05. The data have been replaced in Figure 3b, 3c, Supplementary Figure 1 and Supplementary Table 2 of the revision, and the text have been modified.

2. The authors focus on promoter binding because according to published data (ref 45 and 46) most TFs bind to promoter regions. These references are over 10 years old and since then there has been multiple studies indicating for several TFs that enhancers regulating gene expression are kbs away from the TSS (e.g. ENCODE consortium publications). The authors own data shows that majority of ZNF516 binding is intronic or intergenic not in the promoter region. It is therefore puzzling why the authors are focusing on promoter sites. The authors should indicate why they focused on promoter sites. The argument based on 10-year-old publications is not a strong one.

Authors: Transcription factor binding and the recruitment of RNA Polymerase II to the promoter region is one of the hallmarks of eukaryotic gene transcription. Although transcription factor's association is detected in intronic or intergenic sequences, these sequences are most likely represent enhancer elements which loop back with promoter to enhance transcription; i.e. transcription cannot start from intronic or intergenic regions, a dogma of transcription the reviewer would agree with. This has been further clarified in the revision.

In addition, there seems to be different in the number of promoter sites shown in 3b versus 3c. In 3b there are 5607 ZNF516 promoter sites versus 2993 in 3c (2486+507). Similar difference can be seen with CtBP1. This issue should be clarified.

Authors: Figure 3b showed the number of peaks, while Figure 3c showed the number of genes. We apologize for the confusion and have clarified this in the revision.

3. The authors indicate that: "ChIP-seq samples in our lab are typically subjected to strict quality control by the sequencing company (BGI). Due to the expense and the time, the samples are not replicated.". Replicate sequencing samples are needed to control the biological phenomena not the technical aspects of sequencing. If authors are not willing to perform replicate samples, it should be indicated clearly in the results and in the methods that the results are obtained from a single sequencing sample.

Authors: We appreciate the reviewer for this point and have indicated this in the Methods.

4. Method section lack details of ChIP-seq data analysis. The authors indicate that Bowtie,

MACS2 and CHIPseeker were used without providing any other details. How was the data aligned with Bowtie? How many mismatches were allowed? What is the number of reporter alignments? What parameters were used to build the model in MACS2? Were default settings used? What parameters were used with CHIPseeker?

Authors: The details mentioned above have been added to the Methods.

Other minor issues.

- Figure 4f; what peak in the EGFR promoter represent binding sites assessed in the study? This should be indicated in the figure

Authors: The binding site of ZNF516 on the EGFR promoter has been indicated with a red line in Figure 4f.

- The reference for MACS2 program in the methods is ref 47 which the reference for FIMO.

Authors: The reference for MACS2 program in the Methods has been corrected.

REVIEWERS' COMMENTS:

Reviewer #2 (Remarks to the Author):

NCOMMS-16-24127B

In the 2nd revised manuscript titled "ZNF516 Is a Potent Suppressor of the EGFR Oncogene through Chromatin Targeting of the CtBP/LSD1/CoREST Complex", Sun and colleagues have managed to address my additional comments.